# Synchronized Pruning for Efficient Contrastive Self-Supervised Learning

## Abstract

Various self-supervised learning (SSL) methods have demonstrated strong capability in learning visual representations from unlabeled data. However, the current state-of-the-art (SoTA) SSL methods largely rely on heavy encoders to achieve comparable performance as the supervised learning counterpart. Despite the well-learned visual representations, the large-sized encoders impede the energy-efficient computation, especially for resource-constrained edge computing. Prior works have utilized the sparsity-induced asymmetry to enhance the contrastive learning of dense models, but the generality between asymmetric sparsity and self-supervised learning has not been fully discovered. Furthermore, transferring the supervised sparse learning to SSL is also largely under-explored. To address the research gap in prior works, this paper investigates the correlation between in-training sparsity and SSL. In particular, we propose a novel sparse SSL algorithm, embracing the benefits of contrastiveness while exploiting high sparsity during SSL training. The proposed framework is evaluated comprehensively with various granularities of sparsity, including element-wise sparsity, GPU-friendly $N$:$M$ structured fine-grained sparsity, and hardware-specific structured sparsity. Extensive experiments across multiple datasets are performed, where the proposed method shows superior performance against the SoTA sparse learning algorithms with various SSL frameworks. Furthermore, the training speedup aided by the proposed method is evaluated with an actual DNN training accelerator model.

## 1 Introduction

The early empirical success of deep learning was primarily driven by supervised learning with massive labeled data, e.g., ImageNet (Krizhevsky et al., 2012). To overcome the labeling bottleneck of deep learning, learning visual representations without label-intensive datasets has been widely investigated (Chen et al., 2020a; He et al., 2020; Grill et al., 2020; Zbontar et al., 2021). The recent self-supervised learning (SSL) methods have shown great success and achieved comparable performance to the supervised learning counterpart. The common property of various SSL designs is utilizing different augmentations from the original images to generate contrastiveness, which requires duplicated encoding with wide and deep models (Meng et al., 2022). The magnified training effort and extensive resource consumption make the SSL-trained encoder infeasible for on-device computing (e.g., mobile devices). The contradiction between label-free learning and extraordinary computation cost limits further applications of SSL, also necessitating efficient sparse training techniques for self-supervised learning.

For supervised learning, sparsification (a.k.a. pruning) has been widely explored, aiming to reduce computation and memory costs by removing unimportant parameters during training or fine-tuning. Conventional supervised pruning explores weight sparsity based on a pre-trained model followed by additional fine-tuning to recover the accuracy (Han et al., 2016). For self-supervised learning, recent work (Chen et al., 2021) also sparsified a pre-trained dense SSL model for the downstream tasks with element-wise pruning. In addition to the fine-grained sparsity, MCP (Pan et al., 2022) exploited the filter-wise sparsity on the MoCo-SSL (He et al., 2020) model. Both of these sparse SSL works (Chen et al., 2021; Pan et al., 2022) exploit sparsity based on the pre-trained dense model. However, compared to supervised learning, obtaining the pre-trained model via SSL requires a significant amount of additional training effort (~200 epochs vs. ~1,000 epochs). Therefore, exploring post-training sparsity via fine-tuning is not an ideal solution for efficient SSL.

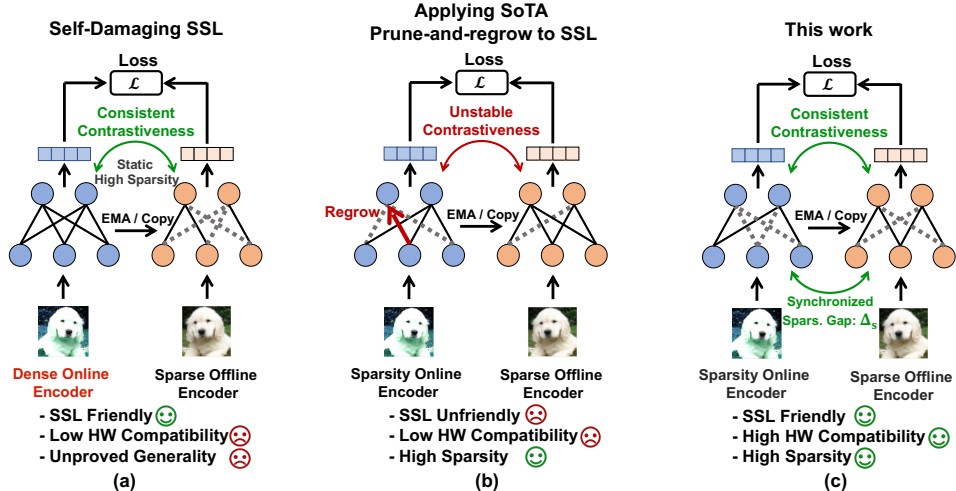

Figure 1: (a) Applying self-damaging scheme (Jiang et al., 2021) to SSL. (b) Applying prune-and-regrow scheme (Liu et al., 2021) to SSL. (c) Proposed contrastiveness-aware sparse training.

On the other hand, sparsifying the model during supervised training (Dettmers & Zettlemoyer, 2019; Evci et al., 2020) has emerged as a promising technique to elevate training efficiency while obtaining a sparse model. To accurately localize the unimportant parameters, prior works investigated various types of important metrics, including gradient-based pruning (Dettmers & Zettlemoyer, 2019) and the "prune-regrow" scheme (Liu et al., 2021). Compared to the post-training sparsification methods, in-training sparsification for supervised training has achieved memory/computation reduction as well as speedup of the training process. However, exploiting in-training sparsity for SSL models that are trained from scratch is still largely under-explored.

More recently, SDCLR (Jiang et al., 2021) proposed the sparsified "self-damaging" encoder, which generates the "sparse-dense" SSL by exploiting fixed high sparsity on one contrastive path (e.g., off-line encoder), while keeping the counterpart dense (e.g., online encoder), as shown in Figure 1(a). Such "sparse-dense" SSL architecture helps to enhance contrastive learning, leading to improved performance with the non-salient and imbalanced samples. Nevertheless, it mainly focuses on the performance enhancement of the SSL-trained dense model (i.e., SimCLR (Chen et al., 2020a)), and whether such a "sparse-dense" asymmetric learning scheme works in other SSL methods remains unclear. On the other hand, the compatibility of the existing SoTA sparse training techniques (Dettmers & Zettlemoyer (2019); Evci et al. (2020); Liu et al. (2021)) is also ambiguous to SSL. As shown in Figure 1(b), they require to frequently prune and regrow the model architecture during SSL training, while SDCLR maintains a fixed "sparse-dense" architecture during the entire training process. The under-explored sparse contrastiveness and expensive self-supervised learning inspire us to investigate the following question: *How to efficiently sparsify the model during self-supervised training with the awareness of contrastiveness?*

To address this question, we present **Sync**hronized **C**ontrastive **P**runing (SyncCP), a novel sparse training algorithm designed for self-supervised learning. To maximize the energy efficiency of SSL training, SyncCP exploits in-training sparsity in both encoders with high compatibility of SSL. The main contributions of this work are:

- We first discover the limitations of the sparsity-induced asymmetric SSL in SDCLR (Jiang et al., 2021) and show that the sparsity-induced "sparse-dense" asymmetric architecture is not universally applicable for various SSL schemes.

- We demonstrate the incompatibility of the SoTA "prune-and-regrow" sparse training method for SSL. Specifically, we formalize the iterative architectural changes caused by applying "prune-and-regrow" to SSL, named as *architecture oscillation,* and observe that frequently updating the pruning candidates lead to larger architecture oscillation, which further hinders the performance of self-supervised learning.

- We present SyncCP, a new sparse training algorithm designed for self-supervised learning. SyncCP gradually exploits high in-training sparsity in both encoders with contrastive

      synchronization and optimally-triggered sparsification, maximizing the training efficiency without hurting the contrastiveness of SSL.

- SyncCP is a general sparse training method for SSL which is compatible with various granularities of sparsity, including element-wise pruning, $N{:}M$ sparsity, and structured sparsity designed for a custom hardware accelerator.
- We validated the proposed method against previous SoTA sparsification algorithms on CIFAR-10, CIFAR-100, and ImageNet datasets. Across various SSL frameworks, SyncCP consistently achieves SoTA accuracy in all experiments.

## 2 RELATED WORKS

### 2.1 CONTRASTIVE SELF-SUPERVISED LEARNING

Self-supervised learning recently has gained popularity due to its ability to learn visual representation without labor-intensive labeling. Specifically, pioneering research works (He et al., 2020; Chen et al., 2020a) utilize the contrastive learning scheme (Hadsell et al., 2006) that aims to group the correlated positive samples while repelling the mismatched negative samples (Oord et al., 2018). The performance of the contrastive learning-based approaches largely depends on the contrastiveness between the positive and negative samples, which requires large-sized batches to support. As indicated by SimCLR (Chen et al., 2020a), the performance of SSL is sensitive to the training batch size, and the inflated batch size elevates training cost. MoCo (He et al., 2020; Chen et al., 2020b) alleviates such issue with queue-based learning and momentum encoder, where the extensive queue-held negative samples provide proficient contrastiveness, and the slow-moving average momentum encoder derives consistent negative pairs. BYOL (Grill et al., 2020) simplifies and outperforms the prior works by only learning positive samples, while the online latent features are projected by an additional predictor $q_\theta$:

$$\text{online prediction} = q_\theta(g_\theta(f_\theta(X))) \tag{1}$$

$$\text{offline target} = g_\xi(f_\xi(X')) \tag{2}$$

Where $f$ and $g$ represent the encoder and projector of online ($\theta$) and offline ($\xi$) paths with augmented input $X$ and $X'$, respectively. The predictor $q_\theta$ generates an alternative view of the projected positive samples, and the offline momentum encoder provides consistent encoding for contrastive learning. Overall, salient and consistent contrastiveness is essential to contrastive self-supervised learning.

### 2.2 SPARSE TRAINING

DNN sparsification has been widely investigated under the supervised learning domain, which can be generally categorized based on the starting point of sparsification. Early works mainly focus on post-training sparsification (Han et al., 2016; Evci et al., 2020; Jayakumar et al., 2020), which removes the weights from the pre-trained model and then recovers the accuracy with subsequent fine-tuning. Other works exploit weight sparsity prior to the training process (Wang et al., 2019; Lee et al., 2018), and the resultant model is trained with the sparsified architecture.

In contrast to post-training or pre-training sparsification, exploiting sparsity during training generates the compressed model with one-time training from scratch, eliminating the requirement of a pre-trained model or extensive searching process. With the full observability of the training process, the magnitude of the gradient can be used to evaluate the model reflection with the exploited sparsity. Motivated by this, GraNet (Liu et al., 2021) utilizes the "prune-and-regrow" technique to periodically remove the unimportant non-zero weights from the sparse model and then regrow certain pruning candidates back. Given the targeted sparsity $s_t$ and total prune ratio $r_t$ at iteration $t$, unimportant weights $w$ are removed based on the Top-K magnitude scores:

$$w^{'} = \text{TopK}(|w|, r_t) \tag{3}$$

Subsequently, the sensitive weights are re-grown back based on the reflection of gradient $g^t$:

$$w = w^{'} + \text{TopK}(g^t_{i!=w', r_t - s_t}) \tag{4}$$

Since the gradient $g_t$ indicates the instant model sensitivity at iteration $t$, the optimal sparse model architecture can be varied between two adjacent pruning steps.

## 2.3 CONTRASTIVE LEARNING WITH SPARSITY-INDUCED ASYMMETRY

As introduced in Section 2.1, salient and consistent contrastiveness is essential for contrastive SSL, where the contrastiveness can be constructed via negative samples or the auxiliary predictors (Grill et al., 2020). Inspired by (Hooker et al., 2019), SDCLR (Jiang et al., 2021) amplifies the contrastiveness by pruning one encoder of SimCLR (Chen et al., 2020a) while keep the identical twin dense. Such "sparsity-induced asymmetry" elevates the performance of SSL with the improved performance of the dense model on the long-tailed data samples. However, SDCLR (Jiang et al., 2021) is not designed for model compression or efficiency improvements. Furthermore, the generality of such sparsity-induced asymmetry remains under-explored in other SSL frameworks.

## 3 CHALLENGE OF SPARSE SELF-SUPERVISED LEARNING

### 3.1 LIMITATIONS OF SPARSITY-INDUCED ASYMMETRY

It has been shown in SDCLR (Jiang et al., 2021) that the sparsity-induced "sparse-dense" asymmetry is beneficial to contrastive SSL. SDCLR (Jiang et al., 2021) is specifically built upon the SimCLR (Chen et al., 2020a) framework with shared encoders, where the pruned architectures have the dense twin in the mirrored contrastive encoder. However, the generality of sparsity-induced asymmetry remains unproved in other SSL methods, which motivates us to investigate the question:

**Question 1:** *For contrastive self-supervised learning with non-identical encoders, will the sparsity-induced asymmetric encoders still result in elevated performance for contrastive learning?*

To answer the above question, we use MoCo-V2 (Chen et al., 2020b) and follow the procedure of SDCLR (Jiang et al., 2021) to generate a highly-sparse online encoder prior to the training process. Given the online and offline (momentum) encoder $\theta$ and $\xi$ with weights $W_\theta$ and $W_\xi$, we have:

$$\text{online output} = g_\theta(f_\theta(X * (\mathbf{M}_\theta \cdot W_\theta))) \tag{5}$$

$$\text{offline output} = g_\xi(f_\xi(X')) \tag{6}$$

The online encoder mask $\mathbf{M}_\theta$ produces a sparse online encoder with initialized element-wise sparsity (Han et al., 2016) at 90%, while the offline encoder is updated by exponential moving average (EMA). The gradient-based update of the online encoder keep recovering the performance drop caused by the high sparsity mask. Following the setup of SDCLR (Jiang et al., 2021), the sparsity is periodically updated at the beginning of each epoch. Table 1 summarizes the linear evaluation accuracy on the CIFAR-10 dataset with different static online sparsity values. As opposed to the performance of SimCLR in (Jiang et al., 2021), directly applying the high sparsity-based perturbation to MoCo-V2 (Chen et al., 2020b) is challenging, and leads to considerable performance degradation. Reversing the sparsity between online and offline encoder also shows the similar results, as presented in Appendix D.

Table 1: Largely degraded performance of MoCo-V2 (Chen et al., 2020b) with self-damaging SSL (Jiang et al., 2021) on CIFAR-10 dataset.

| ResNet-18 | Dense Model Acc. = 92.09% | | | |
|---|---|---|---|---|
| Encoder | Online | Momentum | Online | Momentum |
| Fixed Sparsity | 90% | 0% | 50% | 0% |
| Linear Eval. Acc (%) | 88.72 (-3.41%) | 87.68 (-4.31%) | 92.10 (+0.01%) | 92.07 (-0.02%) |

Summarizing these empirical results, our main observation is:

**Observation 1:** *Compared to the online encoder, the EMA-updated momentum encoder has the delayed architecture, which makes it unqualified to be the "competitor" as SDCLR (Jiang et al., 2021). The directly-applied high sparsity overshoots the asymmetric learning, leading to degraded self-supervised learning.*

### 3.2 FREQUENT ARCHITECTURE CHANGING HINDERS SELF-SUPERVISED LEARNING

As depicted in Eq. 4, the "prune-and-regrow" scheme such as GraNet (Liu et al., 2021) uses instant gradient magnitude to indicate the model sensitivity after magnitude pruning, removing the unimportant and insensitive weights while gradually achieving high sparsity. **Observation 1** demonstrates the incompatibility of the directly-applied high sparsity in SSL, then the following question raises:

Table 2: Sparse training with "prune-and-regrow" scheme on BYOL (Grill et al., 2020).

| BYOL (Grill et al., 2020) | CIFAR-10 Acc (%) | |
|---|---|---|
| ResNet-18 | Dense Model Acc. = 92.42% | |
| Online Encoder Sparsity | 0%→80% | 0%→90% |
| Online Linear Eval. Acc (%) | 91.20±0.02 | 90.13±0.06 |
| Momentum Encoder Sparsity | 0%→50% | 0%→60% |
| Momentum Linear Eval Acc. (%) | 91.31±0.07 | 90.09±0.04 |

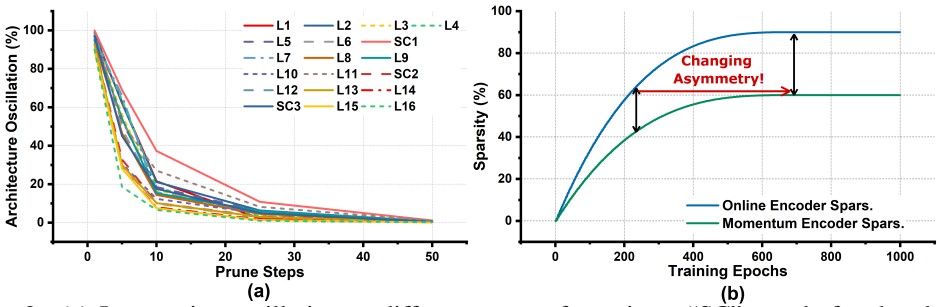

(a)                                 (b)

Figure 2: (a) Layer-wise oscillation at different steps of pruning. "SC" stands for the shortcut connection of ResNet-18 model. (b) Gradually-increased sparsity of GraNet (Liu et al., 2021) leads to inconsistent asymmetry.

**Question 2:** *If we apply the gradually-increased sparsity for both encoders, will the "prune-and-regrow" scheme also be feasible for self-supervised learning?*

To address **Question 2**, we use the SoTA GraNet (Liu et al., 2021) as the example algorithm to exploit in-training sparsity on both encoders of BYOL (Grill et al., 2020), where the regrowing process is only applied to the online encoder. Starting with the dense models, we gradually prune the online and offline encoders to 90% and 60% sparsity in an element-wise fashion with periodically-updated sparsity. For sparse SSL training, the results of such gently-increased sparsity scheme reported in Table 2 outperforms those by (Jiang et al., 2021) (Table 1) by a significant margin. However, the state-of-the-art supervised pruning algorithm still incurs 2.3% linear evaluation accuracy degradation with SSL on the CIFAR-10 dataset.

Compared to the self-damaging SSL with fixed sparsity (Jiang et al., 2021), the "prune-and-regrow" method keeps swapping the pruning candidates to minimize the model sensitivity, oscillating the encoder architecture during training. We quantify such *architecture oscillation* by XORing the masks generated from magnitude pruning $\mathbf{M}_{MP}$ and gradient-based regrow $\mathbf{M}_g$:

$$\mathbf{G}_{cor} = \mathbf{M}_{MP} \oplus \mathbf{M}_g \in \{0, 1\} \tag{7}$$

Under the same sparsity ratio, the number of "1"s in $\mathbf{G}_{cor}$ indicates the amount of architecture oscillation caused by the gradient-based regrow. During the early stage of training, almost all the magnitude pruning candidates are replaced by the regrowing process, as shown in Figure 2(a). The high degree of architecture oscillation implies drastic changes in the sparse model architecture. In the meantime, gradually sparsifying two encoders with different target sparsity further destroys the consistency of self-supervised learning, as shown in Figure 2(b). As a result, we have the following observation for **Question 2**:

**Observation 2:** *Sparsifying the model with frequently changing architecture hinders the contrastiveness and consistency of self-supervised learning and leads to degraded encoder performance.*

As shown in **Observation 1** and **Observation 2**, high sparsity-induced asymmetry is not directly-applicable to sparse SSL, while the consistency requirements of SSL negates the plausibility of gradual sparsity increment. The dilemma between self-supervised learning and sparse training derives the following challenge:

*How can we efficiently sparsify the model during self-supervised training while maximizing the benefits of the sparsity-induced asymmetry?*

## 4 METHOD

To address the above challenge, we propose **Sync**hronized **C**ontrastive **P**runing (SyncCP), which successfully alleviates the contradiction between the needs of high sparsity and the requirements of consistency in self-supervised learning.

### 4.1 SYNCHRONIZED SPARSIFICATION (SYNCS)

The rationale behind the sparsity-induced asymmetric SSL is that the perturbation generated by the pruned encoder elevates the difference between contrastive features. As indicated by **Observation 1** and Table 1, the high sparsity-induced asymmetry is not universally applicable, but the SSL can be rewarded from the asymmetry incurred by lower sparsity (e.g., 50%), where the SSL-trained sparse and dense encoders exhibit negligible accuracy degradation compared to the baseline. Motivated by this, we propose the *Synchronized Sparsification (SyncS)* technique to exploit sparsity in both contrastive encoders. Given the online and offline (momentum) encoder $\theta$ and $\xi$ with weights $W_\theta$ and $W_\xi$, the in-training sparsification can be expressed as:

$$\text{online output} = g_\theta(f_\theta(X * (\mathbf{M}_\theta \cdot W_\theta))) \tag{8}$$

$$\text{offline output} = g_\xi(f_\xi(X' * (\mathbf{M}_\xi \cdot W_\xi))) \tag{9}$$

Where $\mathbf{M}_\theta$ and $\mathbf{M}_\xi$ represent the online and offline (momentum) sparse masks with sparsity $s_\theta$ and $s_\xi$. The proposed SyncS scheme gradually exploits the sparsity in both encoders while maintaining a *consistent sparsity gap* $\Delta_s$ between them during SSL training. At each pruning step $t$, we have:

$$s_\theta^t = s_\theta^f + (s_\theta^i - s_\theta^f)(1 - \frac{t - t_0}{n\Delta t})^3 \tag{10}$$

$$s_\xi^t = s_\xi^f + (s_\xi^i - s_\xi^f)(1 - \frac{t - t_0}{n\Delta t})^3 \tag{11}$$

$$\text{s.t } |s_\theta^t - s_\xi^t| = \Delta_s, \text{ for } t \in \{t_0, t_0 + \Delta t, ..., t_0 + n\Delta t\} \tag{12}$$

The exponent controls the speed of sparsity increment, we adopt the sparsity schedule of Eq. 10-12 from (Liu et al., 2021) to minimize the impact of the parameter tuning. The synchronized sparsity increment with the constraints of $\Delta_s$ prevents the exceeding asymmetry between contrastive encoders while minimizing the distortion caused by the changing sparsity. In practice, $\Delta_s$ is treated as a tunable parameter which impacts the final sparsity of both online and offline encoders. To guarantee the consistency of the contrastive sparsity, both $s_\theta$ and $s_\xi$ are initialized by *Erdos Renyi Kernel* (ERK) (Evci et al., 2020), and with respect to $\Delta_s$, we evaluate the impact of $\Delta_s$ in Appendix.

### 4.2 CONTRASTIVE SPARSIFICATION INDICATOR (CSI)

Achieving high sparsity requires gentle sparsity increment, but as presented in **Observation 2**, the inconsistent architecture difference deteriorates the contrastiveness of SSL. On the other hand, the popular EMA-based update (He et al., 2020) allows the momentum encoder to generate consistent latent representation, but the lagged architecture makes the momentum encoder become an unqualified "competitor" to the online encoder, which violates the findings of (Jiang et al., 2021). To address such conflict, we propose the *Contrastive Sparsification Indicator* (CSI), a simple-yet-effective method that automatically selects the starting point of sparsity increment based on the learning progress of SSL. During the SSL training, CSI first generates the pseudo pruning decisions of both encoders based on element-wise magnitude pruning with target sparsity $s_\theta^f$ and $s_\xi^f$:

$$\mathbf{M}_\theta^* = \mathbb{1}\{|w_\theta| \in \text{TopK}(|w_\theta|, s_\theta^f)\} \tag{13}$$

$$\mathbf{M}_\xi^* = \mathbb{1}\{|w_\xi| \in \text{TopK}(|w_\xi|, s_\xi^f)\} \tag{14}$$

Where $\mathbb{1}$ represents the indicator function, and the resultant pseudo masks of $\mathbf{M}_\theta^*$ and $\mathbf{M}_\xi^*$ will not be applied to the weights. Subsequently, CSI XORs the pseudo pruning masks to generate $\mathbf{G}$ (Eq. 15), and the percentage of "1"s in $\mathbf{G}$ is equivalent to the architecture inconsistency $\mathbf{I}$ (Eq. 16), where $|\mathbf{G}|$ represents the total number of element in $\mathbf{G}$. Instead of using cosine similarity, the bit-wise XOR can be easily implemented on hardware to quantify the architecture difference during training.

$$\mathbf{G} = \mathbf{M}_\theta^* \oplus \mathbf{M}_\xi^* \tag{15}$$

$$\mathbf{I} = 1 - \left(\sum \mathbb{1}\{\mathbf{G} = 0\}\right)/|\mathbf{G}| \tag{16}$$

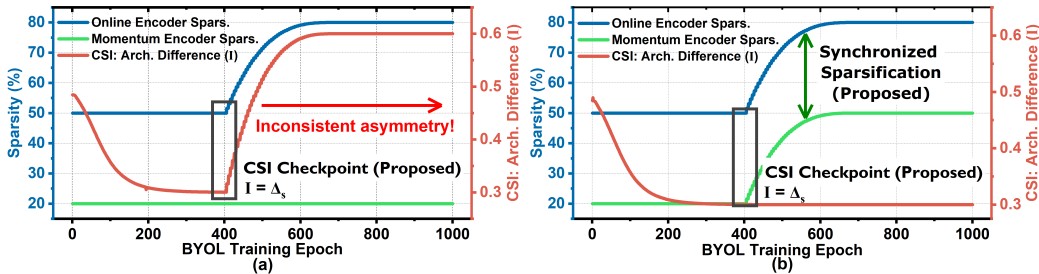

Figure 3: Sparse BYOL training process (a) without SyncS and (b) with SyncS.

Table 3: Performance comparison of BYOL on CIFAR-10 dataset with/without SyncS.

| Method | Online Encoder Spars. | Momentum Encoder Spars. | Online Linear Eval. Acc. (%) |
|---|---|---|---|
| **CSI + SyncS** | 50%→80% | 20%→50% | **92.24%** |
| CSI Only | 50%→80% | 20%→20% (Fixed) | 91.64% |

Given the sparsity gap $\Delta_s$ defined by SyncS, CSI activates the sparsity increment when $\mathbf{I}$ equals to $\Delta_s$, and this moment is defined as the *CSI checkpoint*. In other words, when the architecture difference between online and offline encoders is mainly caused by the sparsity difference, it is the optimal moment to start exploiting the in-training sparsity with the gradually-increased sparsity. With the ability to automatically select the starting point of sparsity increment, the proposed CSI method automatically sparsifies the model with the full awareness of the SSL process. For the SSL framework with shared encoder (Zbontar et al., 2021), the architecture inconsistency $\mathbf{I}$ is computed based on the sparse architecture of two consecutive epochs, and the sparsity increment is activated when $\mathbf{I}$ is less than a pre-defined threshold $\tau$ (e.g., $\tau = 0.1$).

Figure 3 shows the sparsification scheme with and without SyncS. As summarized in Table 3, holding the sparse momentum architecture after the CSI checkpoint interrupts the consistency between the online and momentum encoders. Although the momentum encoder retains the low sparsity at 20%, the absence of consistent asymmetry from synchronized contrastive pruning (SCP) causes the degraded model performance.

On top of the proposed SyncS and CSI schemes, we adopt the prune-and-regrow scheme (Liu et al., 2021) with modifications to exploit sparsity during SSL training. To further alleviate the contrastiveness oscillation caused by changing sparsity, we slowly average the gradient magnitude by exponential moving average (EMA) with gentle momentum, instead of using the instant score. The detailed pseudo code of the proposed algorithm is summarized in Appendix C.

## 5 EXPERIMENTAL RESULTS

In this section, we validate the proposed sparse training scheme and compare it with the current SoTA sparse training schemes. Unlike the work by (Jiang et al., 2021), the proposed scheme exploits in-training sparsity in both contrastive paths (encoders) and aims to achieve energy-efficient self-supervised learning. The proposed method is applied to multiple mainstream SSL frameworks, including EMA-based methods (Chen et al., 2020b; Grill et al., 2020) and SSL with shared encoder (Zbontar et al., 2021). The linear evaluation accuracy and training cost reduction are reported for multiple datasets, including CIFAR-10, CIFAR-100, and ImageNet-2012. Furthermore, this work exploits in-training sparsity with various sparsity granularities, including element-wise sparsity, $N{:}M$ sparsity (Zhou et al., 2020), and structural sparsity for a custom hardware accelerator.

**CIFAR-10 and CIFAR-100** Table 4 summarizes the linear evaluation accuracy of the proposed method on CIFAR-10 and CIFAR-100 datasets with element-wise sparsity. We use ResNet-18 ($1\times$) as the backbone and train the model from scratch by 1,000 epochs. Following the typical high sparsity results reported with supervised learning, we report the model performance with 80% and 90% target sparsity. To sparsify both encoders during SSL training, we initialize the sparsity of online and offline (momentum) encoders as 30% and 0%, where the $\Delta_s$ is set to 30%. The initialized sparse

Table 4: Linear evaluation comparison on CIFAR-10/100 datasets with element-wise sparsity.

| Dataset | | CIFAR-10 Acc (%) | | | CIFAR-100 Acc (%) | | |
|---|---|---|---|---|---|---|---|
| Encoder | | ResNet-18 (1×) | | | ResNet-18 (1×) | | |
| Element-wise Sparsity | | 0% | 80% | 90% | 0% | 80% | 90% |
| MoCo-V2 (Chen et al., 2020b) | **This work** | 92.09 | **91.77±0.08** | **91.31±0.04** | 67.72 | 67.56±0.04 | 66.78±0.07 |
| | GraNet-MoCo (Liu et al., 2021) | 92.09 | 90.66±0.07 | 90.05±0.08 | 67.72 | 67.17±0.05 | 64.92±0.06 |
| | SD-MoCo (Jiang et al., 2021) | 92.09 | 90.26±0.05 | 87.68±0.06 | 67.72 | 65.04±0.04 | 61.33±0.05 |
| BYOL (Grill et al., 2020) | **This work** | 92.42 | **92.26±0.06** | **92.03±0.05** | 68.80 | **68.69±0.06** | **67.73±0.04** |
| | GraNet-BYOL (Liu et al., 2021) | 92.42 | 91.20±0.02 | 90.13±0.03 | 68.80 | 67.17±0.05 | 65.85±0.08 |
| | SD-BYOL (Jiang et al., 2021) | 92.42 | 90.33±0.07 | 87.38±0.04 | 68.80 | 66.13±0.08 | 62.20±0.10 |
| Barlow Twins (Zbontar et al., 2021) | **This work** | 91.74 | **91.67±0.09** | **90.84±0.07** | 68.62 | **68.75±0.13** | **68.48±0.12** |
| | GraNet-Barlow (Liu et al., 2021) | 91.74 | 91.23±0.03 | 90.44±0.12 | 68.62 | 68.40±0.10 | 68.15±0.14 |
| | SD-Barlow (Jiang et al., 2021) | 91.74 | 90.09±0.03 | 88.41±0.07 | 68.62 | 66.42±0.07 | 61.77±0.04 |

Table 5: ImageNet-2012 accuracy and training cost comparison with SoTA works on ResNet-50 with BYOL (Grill et al., 2020).

| ImageNet | | Top-1 Accuracy (%) | FLOPS (Training) | FLOPS (Inference) | Top-1 Accuracy (%) | FLOPS (Training) | FLOPS (Inference) |
|---|---|---|---|---|---|---|---|
| Dense Baseline | | 68.12 | 7.94e+18 (1×) | 8.18e+09 (1×) | 68.12 | 7.94e+18 (1×) | 8.18e+09 (1×) |
| Element-wise Sparsity | | | 80% | | | 90% | |
| Model Size (MB) | | | 20.5 | | | 10.2 | |
| BYOL (Grill et al., 2020) | **This work** | **67.02** | **0.62×** | 0.33 × | **65.67** | **0.56×** | 0.22× |
| | GraNet-BYOL (Liu et al., 2021) | 65.45 | 0.57× | 0.32 × | 64.22 | 0.51× | 0.20× |
| | SD-BYOL (Jiang et al., 2021) | 63.02 | 0.68× | 0.34 × | 60.56 | 0.63× | 0.21× |

The encoders are trained from scratch on the ImageNet dataset with 300 epochs. The FP32 dense ResNet-50 model requires 102MB storage.

encoders reduce the overall memory footprint throughout the entire training process. We rigorously transfer the SoTA GraNet Liu et al. (2021) to SSL based on its open-sourced implementation, the proposed method outperforms GraNet-SSL with 1.26% and 1.86% accuracy improvements on CIFAR-10 and CIFAR-100 datasets, respectively. In all experiments, we report the average accuracy with its variation in 3 runs.

In addition to element-wise sparsity, the recent Nvidia Ampere architecture is equipped with the Sparse Tensor Cores to accelerate the inference computation on GPU with $N{:}M$ structured fine-grained sparsity (Zhou et al., 2020), where the $N$ dense elements remain within an $M$-sized group. Powered by the open-sourced Nvidia-ASP library, SyncCP sparsifies BYOL training (Grill et al., 2020) by targeting 100% $N{:}M$ sparse groups in online encoders. Starting from scratch, the percentage of the $N{:}M$ sparse groups is initialized as 30% and 0% in online and momentum encoders with $\Delta_s$=30%. After the CSI checkpoint, the percentage of $N{:}M$ groups gradually increases. Appendix A describes the detailed pruning algorithm of $N{:}M$ sparsification. Table 6 summarizes linear evaluation accuracy and inference time reduction on the CIFAR-10 and CIFAR-100 datasets. The resultant model achieves up to 2.08× inference acceleration with minimum accuracy degradation. The inference time is measured on an Nvidia 3090 GPU with FP32 data precision.

**ImageNet-2012**   Since the BYOL (Grill et al., 2020) learning scheme achieves the best performance with CIFAR datasets, we further evaluate the proposed method with ResNet-50 on ImageNet based on the BYOL framework (Grill et al., 2020). Following the typical high sparsity results reported in Table 4, we report the model performance with 80% and 90% element-wise sparsity. The data augmentation setup is adopted from the open-sourced library (Costa et al., 2022). Starting from scratch, the model is trained by 300 epochs, where both online and momentum encoders are initialized by ERK with $\Delta_s$ = 30%. While we believe a more fine-grained hyperparameter tuning and extended training efforts could lead to better accuracy, we choose the above scheme for simplicity and reproducibility. Table 5 shows the comparison of linear evaluation accuracy on ImageNet-2012

Table 6: Linear evaluation accuracy comparison on CIFAR-10/100 datasets with $N{:}M$ structured fine-grained sparsity.

| Datasets | CIFAR-10 Acc (%) | | CIFAR-100 Acc (%) | |
|---|---|---|---|---|
| Encoder | ResNet-18 (1×) | | ResNet-18 (1×) | |
| $N{:}M$ Sparse Pattern | 2:4 | 1:4 | 2:4 | 1:4 |
| MoCo-V2 (Chen et al., 2020b) | 91.99±0.07 | 91.53±0.04 | 67.58±0.05 | 67.11±0.05 |
| BYOL (Grill et al., 2020) | 92.61±0.05 | 91.83±0.02 | 68.69±0.02 | 68.09±0.07 |
| Barlow Twins (Zbontar et al., 2021) | 91.68±0.04 | 90.97±0.03 | 68.26±0.07 | 68.19±0.06 |
| Inference time reduction (s) | 1.40× | 2.08× | 1.40× | 2.08× |
| FLOPs (Dense = 5.56e+8) | 0.50× | 0.25× | 0.50× | 0.25× |
| Weight Memory (MB) | 22.4 | 11.2 | 22.4 | 11.2 |

The FP32 dense ResNet-18(1×) model requires 44.8MB storage and takes 1.27 seconds per 10K testset images during inference.

dataset. Compared to the self-damaging scheme (Jiang et al., 2021) and GraNet (Liu et al., 2021), the proposed algorithm achieves the same highly-sparse network with 4.72% and 1.21% Top-1 inference accuracy improvements, respectively. GraNet exploits in-training sparsity throughout the entire training process, but the inconsistent contrastiveness hampers the model performance. On the other hand, the dense encoder limits the efficiency of the self-damaging scheme (Jiang et al., 2021) scheme, and the static high sparsity degrades the model performance. It has been shown that SSL-trained encoders are strong visual learners (Grill et al., 2020; Ericsson et al., 2021). Appendix A summarizes the performance of the proposed algorithm by fine-tuning the ImageNet-trained sparse encoders on CIFAR-10 and CIFAR-100 datasets. With only 300 epochs of sparse SSL training, the resultant sparse encoder outperforms the SoTA supervised sparse training algorithms.

Table 7: Hardware training acceleration of the proposed structured SyncCP on CIFAR-10 datasaet.

| BYOL+ResNet-18 | | Top-1 Accuracy (%) | Training Speed-up | Top-1 Accuracy (%) | Training Speed-up |
|---|---|---|---|---|---|
| Dense Baseline | | 92.42 | 1× | 92.42 | 1× |
| Target Structured Sparsity | | 80% | | 90% | |
| BYOL (Grill et al., 2020) | This work | 92.16 | 1.74× | 91.77 | 1.91× |

**Hardware-based Structured Pruning** The hardware practicality of element-wise sparsification is often limited by the irregularity of fine-grained sparsity and index requirement. To that end, we employ structured sparsity based on group-wise EMA scores towards achieving actual hardware training acceleration. The encoders are initialized by ERK with 30% and 0% sparse groups while keeping $\Delta_s = 30\%$. The structured sparsity starts to gradually increase after the CSI checkpoint. We adopt the training accelerator specifications from (Venkataramanaiah et al., 2022) and choose $K_l$ (# of output channels) $\times C_l$ (# of input channels) = 8×8 as the sparse group size. Table 7 evalautes the training speedup of BYOL (Grill et al., 2020) aided by the structured sparse training. The proposed algorithm achieves up to 1.91× training acceleration with minimal accuracy degradation.

## 6 CONCLUSION

In this paper, we propose a novel sparse training algorithm designed for self-supervised learning (SSL). As one of the first studies in this area, we first point out the imperfections of the sparsity-induced asymmetric self-supervised learning, as well as the incompatibility of the supervised sparse training algorithm in SSL. Based on the well-knit conclusions, we propose a contrastiveness-aware sparse training algorithm, consisting of synchronized contrastive pruning (SCP) and contrastive sparsification indicator (CSI). The proposed method outperforms the SoTA sparse training algorithm on both CIFAR and ImageNet-2012 datasets with various mainstream SSL frameworks. We also demonstrate the actual training and inference hardware acceleration with structured sparsity and $N{:}M$ structured fine-grained pattern.

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

# A DOWNSTREAM TASKS PERFORMANCE WITH THE PRE-TRAINED SPARSE ENCODER

The pre-trained sparse encoder can be used for downstream tasks. We verified the performance sparse ImageNet-trained model in Table 5 with the downstream tasks. The following table summarizes the transfer learning performance by fine-tuning the ImageNet-trained sparse BYOL encoder on CIFAR-10 and CIFAR-100 datasets.

Table 8: Transfer learning performance of the ImageNet-trained BYOL encoder on CIFAR-10 and CIFAR-100 datasets.

| Dataset | | CIFAR-10 Acc (%) | | CIFAR-100 Acc. (%) | |
|---|---|---|---|---|---|
| **Encoder** | | **ResNet-50 (1×)** | | **ResNet-50 (1×)** | |
| **Element-wise Sparsity** | | 0% | 90% | 0% | 90% |
| Supervised Training | SNIP (Lee et al., 2018) | 94.75 | 92.65 | 78.23 | 73.14 |
| | GraSP (Wang et al., 2019) | 94.75 | 92.47 | 78.23 | 73.28 |
| | RigL (Evci et al., 2020) | 94.75 | 94.45 | 78.23 | 76.50 |
| | GraNet (Liu et al., 2021) | 94.75 | 94.64 | 78.23 | 77.89 |
| Transfer learning from BYOL-trained model | **This work** | 96.09 | **95.72** | 80.13 | **79.48** |

The pre-trained sparse encoder are consistently preserved during the downstream fine-tuning. As shown in the table above, the strong visual representation learners are obtained by training the backbone model with self-supervised learning on ImageNet dataset. Sparsified by the proposed algorithm on ImageNet with 300 epochs during pre-training, fine-tuning the SSL-trained backbone model leads to 1.08% and 1.59% accuracy improvements compared to the recent SoTA supervised pruning algorithms.

# B THE IMPACT OF $\Delta_s$

We evaluate the impact of $\Delta_s$ with different sparsification schemes on CIFAR-10 dataset with BYOL SSL framework. With the target sparsity = 90%, the following table summarizes the accuracy and training cost reduction with different $\Delta_s$ values associated with different initial and final density.

Table 9: The linear evaluation accuracy and the training FLOPs reduction with sparsity gap $\Delta_s$.

| $\Delta_s$ | Online Encoder Spars. | Offline Encoder Spars. | Online Linear Eval Acc. | FLOPs (Training) |
|---|---|---|---|---|
| 50% | 50% → 90% | 0% → 40% | 91.68 | 0.60× |
| 40% | 50% → 90% | 10% → 50% | 91.41 | 0.58× |
| 30% | 50% → 90% | 20% → 60% | 91.27 | 0.53× |
| 40% | 40% → 90% | 0% → 50% | **91.79** | **0.57×** |
| 30% | 30% → 90% | 0% → 60% | **92.02** | **0.56×** |

Given the fixed target sparsity = 90%, it is easy to tell that the higher $\Delta_s$ leads to denser momentum encoder and less computation reduction. Meanwhile, sparsifies the momentum encoder at the beginning of training is sub-optimal. Furthermore, less initial sparsity and smaller $\Delta_s$ value (4th row) achieves the best tradeoff between computation reduction and model performance.

## C  APPENDIX C

### C.1  PSEUDO CODE OF SYNCCP WITH ELEMENT-WISE SPARSITY

---

**Algorithm 1:** Synchornized Contrastive Pruning (SyncCP)

---

**Initialize** Sparse online encoder $f_\theta$, Sparse offline encoder $f_\xi$, EMA updater, Momentum $\gamma$,
**SyncS** density gap $\Delta_s$, **CSI** threshold $\tau$ (Default=$\Delta_s$).
Initial sparsity $s_\theta^0$, $s_\xi^0$, such that $|s_\theta^* - s_\xi^*| = \Delta_s$
Target sparsity $s_\theta^*$, $s_\xi^*$, such that $|s_\theta^* - s_\xi^*| = \Delta_s$
Initial mask $\mathbf{M}_\theta^0$, $\mathbf{M}_\xi^0$.
Pruner udpate frequency $\Phi$
**while** $t <$ *Total Iterations* **do**
    Draw augmented data $(X, X')$;
    **Forward pass:** online encoding $= f_\theta(\mathbf{M}_\theta \cdot \theta, X)$ ;
    **Forward pass:** offline encoding $= f_\xi(\mathbf{M}_\xi \cdot \xi, X')$;
    **Update** Exponential Moving Average (**EMA**) gradient score based on Eq. 17 ;
    **if** *End Epoch* **then**
        Get pseudo masks $\mathbf{M}_\theta^*$ and $\mathbf{M}_\xi^*$ based on magnitude pruning;
        Compute layer-wise $\mathbf{G}$ and $\mathbf{I}$ based on Eq. 15 and Eq. 16;
        **if** $\mathbf{I} = \Delta_s$ **then**
            | *Prune*=True
        **end**
    **end**
    **if** $t \% \Phi = 0$ **then**
        **if** *Prune=True* **then**
            **Update** sparsity $s_\theta^t$, $s_\xi^t$ based on Eq. 10 and Eq. 11;
            Maintain the **SyncS** constraint $\Delta_s$;
            Inside $f_\theta$ and $f_\xi$, prune $s_\theta^t$, and $s_\xi^t$ elements with least magnitude score;
            Prune extra $r_\theta^t$ elements of the unpruned elements, then regrow $r_\theta$ elements with
             hights EMA-gradient score;
            **Update** $\mathbf{M}_\theta^t$, $\mathbf{M}_\xi^t$ based on Eq. 3 and Eq. 4;
        **else**
        **end**
    **end**
**end**

---

### C.2  EMA-BASED PRUNE AND REGROW

As aforementioned, the findings of **Observation 2** implies the incompatibility of the instant gradient and magnitude score. Together with the proposed SyncS and CSI methods, weight importance is measured by the magnitude score, while the sensitivity of the model is quantified by the gently averaged gradient magnitude with EMA:

$$\bar{g}^t = \gamma \times \bar{g}^{t-1} + (1 - \gamma) \times |g|^t \tag{17}$$

Table 10 summarizes the linear evaluation accuracy of ResNet-18 trained by BYOL (Grill et al., 2020). We initialize $s_\theta^0$ and $s_\xi^0$ as 40% and 10%, where the $\Delta_s$ is set to 30%, the EMA momentum is set to 0.1 for gentle gradient score averaging.

| Metric | Prune | Regrow | EMA | Online Linear Eval. Acc. (%) |
|---|:---:|:---:|:---:|:---:|
| **This work** | ✓ | ✓ | ✓ | **91.88** |
| Prune-and-regrow | ✓ | ✓ | ✗ | 91.52 |
| Magnitude Pruning | ✓ | ✗ | ✗ | 90.99 |

Table 10: Peformance comparison between different sparsification metrics.

### C.3 PSEUDO CODE OF SYNCCP WITH $N{:}M$ SPARSITY

---

**Algorithm 2:** Synchornized Contrastive Pruning (SyncCP) with $N{:}M$ Sparsity

---

**Initialize** Sparse online encoder $f_\theta$, Sparse offline encoder $f_\xi$, EMA updater, Momentum $\gamma$,
**SyncS** density gap $\Delta_s$, **CSI** threshold $\tau$ (Default=$\Delta_s$).
**Group size $M$, Number of dense element per group $N$.**
Initial percentage $p_\theta^0$ of $N{:}M$ groups in $f_\theta$, Initial percentage $p_\xi^0$ of $N{:}M$ groups in $f_\xi$;
Such that $|p_\theta^0 - p_\xi^0| = \Delta_s$;
Target percentage $p_\theta^* = 100\%$, $p_\xi^* = p_\theta^* - \Delta_s$;
Initial mask $\mathbf{M}_\theta^0$, $\mathbf{M}_\xi^0$.
Pruner udpate frequency $\Phi$.
**while** $t < $ *Total Iterations* **do**
> Draw augmented data $(X, X')$;
> **Forward pass:** online encoding $= f_\theta(\mathbf{M}_\theta \cdot \theta, X)$ ;
> **Forward pass:** offline encoding $= f_\xi(\mathbf{M}_\xi \cdot \xi, X')$;
> **Update** Exponential Moving Average (**EMA**) weight gradient score based on Eq. 17 ;
> **if** *End Epoch* **then**
>> Get pseudo masks $\mathbf{M}_\theta^*$ and $\mathbf{M}_\xi^*$ based on magnitude pruning;
>> Compute layer-wise $\mathbf{G}$ and $\mathbf{I}$ based on Eq. 15 and Eq. 16;
>> **if** $\mathbf{I} = \Delta_s$ **then**
>>> *Prune*=True
>>
>> **end**
>
> **end**
> **if** $t \% \Phi = 0$ **then**
>> **if** *Prune=True* **then**
>>> **Update** sparsity $p_\theta^t$, $p_\xi^t$ based on Eq. 10 and Eq. 11;
>>> Maintain the **SyncS** constraint $p_\theta^t$, $p_\xi^t = p_\theta^t - \Delta_s$;
>>> Inside $f_\theta$ and $f_\xi$, pick $p_\theta^t$, and $p_\xi^t$ M-sized groups with least sum of magnitude score;
>>> Inside each group, prune the N-M elements with smallest magnitude score;
>>> **Update** $\mathbf{M}_\theta^t$, $\mathbf{M}_\xi^t$ based on Figure 4;
>>
>> **else**
>> **end**
>
**end**

---

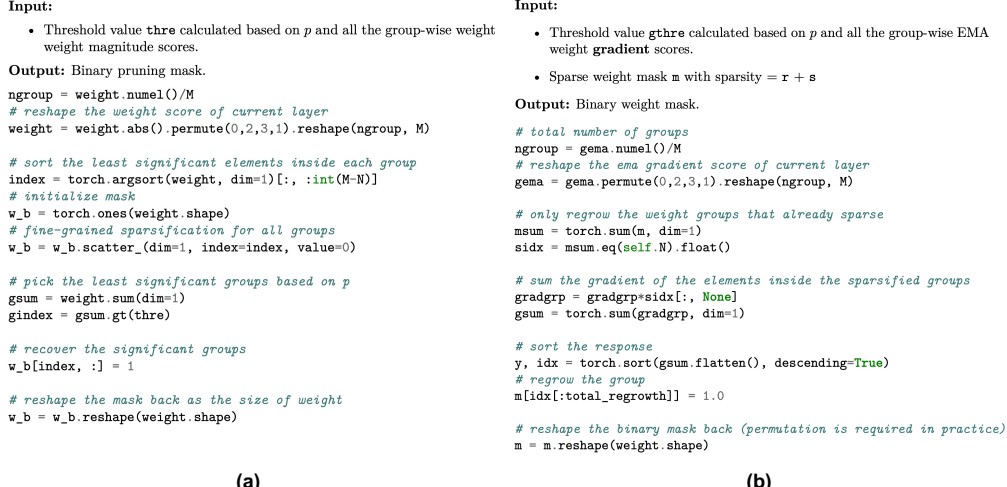

Figure 4: Group-wise (a) prune and (b) regrow algorithm based on EMA gradient score. SyncCP sparsifies $M - N$ elements inside each group, while keep $p^t$ $N{:}M$ groups inside $f$.

## C.4 PSEUDO CODE OF SYNCCP WITH STRUCTURED SPARSITY

---

**Algorithm 3:** Synchornized Contrastive Pruning (SyncCP) with Structured Sparsity

---

**Initialize** Sparse online encoder $f_\theta$, Sparse offline encoder $f_\xi$, EMA updater, Momentum $\gamma$,
**SyncS** density gap $\Delta_s$, **CSI** threshold $\tau$ (Default=$\Delta_s$).
**Group size** $K_l$ (**# of output channels**) $\times C_l$ (**# of input channels**) = $g \times g$**.**
Initial percentage $p_\theta^0$ of sparse groups in $f_\theta$, Initial percentage $p_\xi^0$ of sparse groups in $f_\xi$;
Such that $|p_\theta^* - p_\xi^*| = \Delta_s$;
Initial structured sparsity $s_\theta^0$, $s_\xi^0$, such that $|s_\theta^* - s_\xi^*| = \Delta_s$
Target structured sparsity $s_\theta^*$, $s_\xi^*$, such that $|s_\theta^* - s_\xi^*| = \Delta_s$
Initial mask $\mathbf{M}_\theta^0$, $\mathbf{M}_\xi^0$.
Pruner udpate frequency $\Phi$.
**while** $t <$ *Total Iterations* **do**
    Draw augmented data $(X, X')$;
    **Forward pass:** online encoding $= f_\theta(\mathbf{M}_\theta \cdot \theta, X)$ ;
    **Forward pass:** offline encoding $= f_\xi(\mathbf{M}_\xi \cdot \xi, X')$;
    **Update** Exponential Moving Average (**EMA**) gradient score based on Eq. 17;
    **if** *End Epoch* **then**
        Get pseudo masks $\mathbf{M}_\theta^*$ and $\mathbf{M}_\xi^*$ based on magnitude pruning;
        Compute layer-wise $\mathbf{G}$ and $\mathbf{I}$ based on Eq. 15 and Eq. 16;
        **if** $\mathbf{I} = \Delta_s$ **then**
            *Prune*=True
        **end**
    **end**
    **if** $t \% \Phi = 0$ **then**
        **if** *Prune=True* **then**
            **Update** structured sparsity $s_\theta^t$, $s_\xi^t$ based on Eq. 10 and Eq. 11;
            Maintain the **SyncS** constraint $s_\theta^t, s_\xi^t = s_\theta^t - \Delta_s$;
            Inside $f_\theta$ and $f_\xi$, pick $s_\theta^t$, and $s_\xi^t$ groups with least sum of magnitude score;
            Outside the sparsified groups of $f_\theta$, prune $r_\theta^t$ more groups with least sum of
              magnitude score;
            Among the sparsified groups $f_\theta$, regrow the $r_\theta^t$ groups back with highest sum of
              EMA gradient score;
            **Update** $\mathbf{M}_\theta^t$, $\mathbf{M}_\xi^t$;
        **end**
    **end**
**end**

---

# D SPARSIFYING ONLINE OR OFFLINE ENCODERS

We mirror the experiment of Table 1 by exploiting high sparsity in the momentum encoder while keeping the online encoder dense:

Table 11: Largely degraded performance of MoCo-V2 with self-damaging SSL on CIFAR-10 dataset.

| ResNet-18 | Dense Model Acc. = 92.09% | | | |
|---|---|---|---|---|
| **Encoder** | **Online** | **Momentum** | **Online** | **Momentum** |
| Fixed Sparsity | 90% | 0% | 50% | 0% |
| Linear Eval. Acc (%) | 88.72 (-3.41%) | 87.68 (-4.31%) | 92.10 (+0.01%) | 92.07 (-0.02%) |
| Fixed Sparsity | 0% | 90% | 0% | 50% |
| Linear Eval. Acc (%) | 88.44 (-3.65%) | 87.52 (-4.57%) | – | – |

With the proposed CSI + SyncS sparsification method, we empirically observe that exploiting high sparsity in online model leads to better performance with gradient-based model update. Table 12 summarizes the comparison results of BYOL on CIFAR-10 dataset with the proposed algorithm.

Table 12: Performance comparison of exploiting higher sparsity in online or momentum encoders.

| Method | Online Encoder Spars. | Momentum Encoder Spars. | Online Linear Eval. Acc. (%) |
|---|---|---|---|
| **CSI + SyncS** | 0%→50% | 30%→80% | 91.88% |
| **CSI + SyncS** | 30%→80% | 0%→50% | **92.24%** |

Overall, sparsifying the online encoder leads to the optimal performance.

# E   DETAILED EXPERIMENTAL SETUP OF SYNCCP

## E.1   LINEAR EVALUATION PROTOCOL

As in (Kolesnikov et al., 2019; Kornblith et al., 2019; Chen et al., 2020a), we use the standard linear evaluation protocol on CIFAR-10/100 and ImageNet-2012 datasets, which training a linear classifier on top of the frozen SSL-trained encoder. During linear evaluation, we apply spatial augmentation and random flips. The linear classifier is optimized by SGD with cross-entropy loss.

## E.2   CIFAR-10/100 EXPERIMENTS

The training hyper-parameters of the compared individual sparse training works are same for CIFAR-10 and CIFAR-100. We provide the detailed training setup of different self-supervised learning frameworks as follow:

**MoCo-V2**  The ResNet-18 ($\times$) encoder is trained by MoCo-V2 (Chen et al., 2020b) from scratch by 1,000 epochs with SGD optimizer and 256 batch size. The learning rate is set to 0.3 with Cosine learning rate decay and 10 epochs warmup. The detailed data augmentation is summarized in Table 13.

| Parameter | $X$ | $X'$ |
|---|---|---|
| Random crop size | $32 \times 32$ | $32 \times 32$ |
| Horizontal flip probability | 0.5 | 0.5 |
| Color jitter probability | 0.8 | 0.8 |
| Brightness adjustment probability | 0.4 | 0.4 |
| Contrast adjustment probability | 0.4 | 0.4 |
| Saturation adjustment probability | 0.2 | 0.2 |
| Hue adjustment probability | 0.1 | 0.1 |
| Gaussian blurring probability | 0.0 | 0.0 |
| Solarization probability | 0.0 | 0.0 |

Table 13: Detailed image augmentation settings for MoCo-V2 (Chen et al., 2020b) on CIFAR-10/100.

**BYOL**  The ResNet-18 ($\times$) encoder is trained by BYOL (Grill et al., 2020) from scratch by 1,000 epochs with LARS-SGD optimizer (You et al., 2017). The predictor is constructed with 4096 hidden features and 256 output dimension. We use 256 batch size along with 1.0 learning rate. The Cosine learning rate scheduler is used with 10 epochs warmup training. The detailed data augmentation is summarized in Table 14.

| Parameter | $X$ | $X'$ |
|---|---|---|
| Random crop size | $32 \times 32$ | $32 \times 32$ |
| Horizontal flip probability | 0.5 | 0.5 |
| Color jitter probability | 0.8 | 0.8 |
| Brightness adjustment probability | 0.4 | 0.4 |
| Contrast adjustment probability | 0.4 | 0.4 |
| Saturation adjustment probability | 0.2 | 0.2 |
| Hue adjustment probability | 0.1 | 0.1 |
| Gaussian blurring probability | 0.0 | 0.0 |
| Solarization probability | 0.0 | 0.2 |

Table 14: Detailed image augmentation settings for BYOL (Chen et al., 2020b) on CIFAR-10/100.

**Barlow Twins** The ResNet-18 ($\times$) encoder is trained by Barlow Twins (Zbontar et al., 2021) from scratch by 1,000 epochs with LARS-SGD optimizer (You et al., 2017). We use 256 batch size along with 0.3 learning rate and $1e-4$ weight decay. The Cosine learning rate scheduler is used with 10 epochs warmup training. The detailed data augmentation is summarized in Table 15.

| Parameter | $X$ | $X'$ |
|---|---|---|
| Random crop size | $32 \times 32$ | $32 \times 32$ |
| Horizontal flip probability | 0.5 | 0.5 |
| Color jitter probability | 0.8 | 0.8 |
| Brightness adjustment probability | 0.4 | 0.4 |
| Contrast adjustment probability | 0.4 | 0.4 |
| Saturation adjustment probability | 0.2 | 0.2 |
| Hue adjustment probability | 0.1 | 0.1 |
| Gaussian blurring probability | 0.0 | 0.0 |
| Solarization probability | 0.0 | 0.2 |

Table 15: Detailed image augmentation settings for Barlow Twins (Zbontar et al., 2021) on CIFAR-10/100.

### E.3 IMAGENET EXPERIMENTS

Starting from scratch, the proposed SyncCP algorithm exploits in-training sparsity with the BYOL framework (Grill et al., 2020) on ImageNet-2012 dataset. The ResNet-50 encoder is trained by LARS-SGD (You et al., 2017) with 0.45 learning rate and a momentum of 0.9. We uses 0.1 for the for EMA-averaged gradient score. We use 128 batch size along with 1e-6 weight decay. The detailed image augmentations are summarized in Table 16.

| Parameter | $X$ | $X'$ |
|---|---|---|
| Random crop size | $224 \times 224$ | $224 \times 224$ |
| Horizontal flip probability | 0.5 | 0.5 |
| Color jitter probability | 0.8 | 0.8 |
| Brightness adjustment probability | 0.4 | 0.4 |
| Contrast adjustment probability | 0.4 | 0.4 |
| Saturation adjustment probability | 0.2 | 0.2 |
| Hue adjustment probability | 0.1 | 0.1 |
| Gaussian blurring probability | 1.0 | 0.1 |
| Solarization probability | 0.0 | 0.2 |

Table 16: Detailed image augmentation settings for BYOL (Grill et al., 2020) on ImageNet.

# F  COMPUTATION REDUCTION WITH DIFFERENT SPARSITY VALUES

Table 17: ImageNet-2012 linear evaluation accuracy and training cost comparison with different sparsity on ResNet-50 with BYOL (Grill et al., 2020).

| Dataset | ImageNet-2012 | | |
|---|---|---|---|
| Sparsity | Top-1 Accuracy (%) | FLOPs (Training) | FLOPs (Inference) |
| 0% | 68.12 | 7.94e+18 (1×) | 8.18e+09 (1×) |
| 50% | 68.31 | 0.78× | 0.68× |
| 80% | 67.02 | 0.62× | 0.33× |
| 90% | 65.67 | 0.56× | 0.22× |

