# OpenReview forum: "Synchronized Contrastive Pruning for Efficient Self-Supervised Learning"
_ICLR.cc/2023/Conference — Submitted to ICLR 2023_

### Official Review · Reviewer_EBsm · 2022-10-24

**Confidence:** 2
**Correctness:** 3
**Technical Novelty And Significance:** 2
**Empirical Novelty And Significance:** 2
**Recommendation:** 5

**Clarity, Quality, Novelty And Reproducibility:**

Clarity could be improved:

1.  Eq 10-12: Some of the variables and superscripts could be defined.  What is the motivation for the form in Eq 10 and 11, particularly the cubic exponent?

2.  What is the difference between the two activations: "sparsity increment" and "sparsification process?"   They seem to have different conditions to be activated.    Why is τ independent of $\Delta_s$?

"CSI activates the sparsity increment when I equals to ∆ s , and this moment is defined as the CSI checkpoint"

"the sparsification process is activated when I is less than a pre-defined threshold τ (e.g., τ = 0.9)


Reproducibility could be affected by some parts that could use further explanation.

**Strength And Weaknesses:**

Strengths:

1.  They compared SyncCP with two existing sparsification approaches on 3 datasets.  Generally, the results indicate that SyncCP compares favorably.

2.  Motivation on the disadvantage of sparsification inconsistency in asymmetric encoders in SSL.

Weaknesses:

1.  Some parts of the approach could be explained further (see next section)

2.  An analysis of how $\Delta_s$ affect performance could be studied.

3.  Additional downstream tasks would help illustrate the quality of learned representations.

**Summary Of The Paper:**

For self-supervised learning with contrastive loss, they proposed Synchronized Contrastive Pruning (SyncCP) that have two encoders (online and offline/momentum) with different sparsity, but the difference in sparsity is maintained at $\Delta_s$.  They also propose Contrastive Sparsification Index (CSI) that determines when to start sparsification.  CSI calculates the difference in the pruning masks (M) as inconsistency (I).  When inconsistency reaches the sparsity gap $\Delta_s$, CSI activates the sparsity increment.

They compare their proposed approach with two existing sparsification approaches using 3 SSL approaches (MoCo, BYOL, Barlow Twins) and 2 datasets (CIFAR 10 and 100).   With Imagenet, they only use BYOL in SSL.   Generally, the results indicate that the proposed approach compares favorably.



**Summary Of The Review:**

The authors propose SyncCP to synchronize sparsity in asymmetric encoders in SSL.   SnycCP constraint the difference in sparsity to $\Delta_s$.   Empirical results indicate SyncCP compares favorably with 2 other sparsification approaches.  Additional analysis on $\Delta_s$ and downstream tasks can benefit the paper.

---

> ### Author Response · Authors · 2022-11-19
> **Response to Reviewer EBsm: Q1-Q3**
>
> **Q1:**  Some parts of the approach could be explained further (see next section)
>
> **Q2:** An analysis of how $\Delta_s$ affect performance could be studied.
>
> **A2:** Thank you for your question. We evaluate the impact of $\Delta_s$ with different sparsification schemes on CIFAR-10 dataset with BYOL SSL framework. With the target sparsity = 90%, the following table summarizes the accuracy and training cost reduction with different $\Delta_s$ values associated with different initial and final density.
>
> | $\Delta_s$ | Online Encoder Spars. | Momentum Encoder Spars. | Online Linear Eval. Acc. (%) | FLOPS (Training) |
> | :--------: | :-------------------: | :---------------------: | :--------------------------: | :--------------: |
> |    50%     |  50%$\rightarrow$90%  |   0%$\rightarrow$40%    |            91.68             |   0.60$\times$   |
> |    40%     |  50%$\rightarrow$90%  |   10%$\rightarrow$50%   |            91.41             |   0.58$\times$   |
> |    30%     |  50%$\rightarrow$90%  |   20%$\rightarrow$60%   |            91.27             |   0.53$\times$   |
> |    40%     |  40%$\rightarrow$90%  |   0%$\rightarrow$50%    |            91.79             |   0.57$\times$   |
> |    30%     |  30%$\rightarrow$90%  |   0%$\rightarrow$60%    |          **92.02**           | **0.56**$\times$ |
>
> Given the fixed target sparsity of 90%, it is easy to tell that the higher $\Delta_s$ leads to denser momentum encoder and less computation reduction.
> Meanwhile, sparsifying the momentum encoder at the beginning of training is sub-optimal.  Furthermore, less initial sparsity and smaller $\Delta_s$ value (the bottom row) achieves the best tradeoff between computation reduction and model performance. We added the above table in the Appendix of the revised manuscript as Table 9.
>
> **Q3:** Additional downstream tasks would help illustrate the quality of learned representations.
>
> **A3:** Thank you for your question. We verified the performance sparse ImageNet-trained model with the downstream tasks. The following table summarizes the transfer learning performance by fine-tuning the ImageNet-trained sparse BYOL encoder on CIFAR-10 and CIFAR-100 datasets.
>
> *[Table: Transfer learning performance of the ImageNet-trained BYOL encoder on CIFAR-10 and CIFAR-100 datasets.]*
>
> |                  Dataset                   |               | CIFAR-10 Acc (%) | CIFAR-10 Acc (%) | CIFAR-100 Acc (%) | CIFAR-100 Acc (%) |
> | :----------------------------------------: | :-----------: | :--------------: | :--------------: | :---------------: | :---------------: |
> |                  Encoder                   |               |    ResNet-50     |    ResNet-50     |     ResNet-50     |     ResNet-50     |
> |           Element-wise Sparsity            |               |        0%        |       80%        |        0%         |        90%        |
> |                                            |   SNIP [R1]   |      94.75       |      92.65       |       78.23       |       73.14       |
> |                                            |  GraSP [R2]   |      94.75       |      92.47       |       78.23       |       73.28       |
> |            Supervised Training             |   RigL [R3]   |      94.75       |      92.47       |       78.23       |       73.28       |
> |                                            |  GraNet [R4]  |      94.75       |      94.45       |       78.23       |       76.50       |
> | Transfer learning  from BYOL-trained model | **This work** |      96.09       |    **95.72**     |       80.13       |     **79.48**     |
>
> Please note that the pre-trained sparse architecture are consistently preserved during the downstream fine-tuning.
> As shown in the table above, the strong visual representation learners are obtained by training the backbone model with self-supervised learning on ImageNet dataset. Sparsified by the proposed algorithm on ImageNet with 300 epochs during pre-training, fine-tuning the SSL-trained backbone model leads to 1.08% and 1.59% accuracy improvements compared to the recent SoTA supervised pruning algorithms. We added the table above in the Appendix of the revised manuscript as Table 8.

---

> > ### Author Response · Authors · 2022-11-19
> > **Response to Reviewer EBsm: Clarity and Quality**
> >
> > - Eq 10-12: Some of the variables and superscripts could be defined. What is the motivation for the form in Eq 10 and 11, particularly the cubic exponent?
> >
> >   - **Response: ** The exponent of Eq.~(10)-(12) tunes the speed of the sparsity increment during SSL training. The cubic exponent is selected by the recent SoTA sparse training method [R4]. We adopted the same hyperparameter selection in this work to minimize the impact of hyperparameter tuning. In Section 4.1 of the revised manuscript, we clarify the definition of variables and sparsity schedule.
> >
> > - What is the difference between the two activations: "sparsity increment'' and "sparsification process?'' They seem to have different conditions to be activated. Why is $\tau$ independent of $\Delta_s$?
> >
> >   - **Response:** "Sparsification'' and "sparsity increment'' all refer to increasing model sparsity. To avoid the further confusion, we have unified all the obfuscate descriptions of "sparsification'' to "sparsity increment'' in Section 3 and Section 4 of the revised manuscript.
> >
> >   - **Regarding the relationship between $\tau$ and $\Delta_s$:**
> >
> >   - $\tau$ is the inconsistency threshold designed for the self-supervised learning method with shared encoders. For the SSL frameworks with online and offline (momentum) encoders, the $\mathbf{I}$ measures the architecture inconsistency between online and offline encoders. 1) For the SSL training scheme with shared encoders, the proposed SyncCP holds the sparsity difference $\Delta_s$ between the contrastive encoding, when $\mathbf{I}$ and $\Delta_s$ reaches equality, the contrastiveness between two encoders reaches consistency. 2) For the shared encoder-based SSL, measuring the architecture difference between two consecutive epochs is a simple-yet-effective method to justify the proper starting point to increase the sparsity. For consecutive epochs $e$ and $e+1$, the architecture difference is measured as: $G = M_e^* \oplus M_{e+1}^*$
> >
> > And the architecture inconsistency is:
> >
> > $\mathbf{I} = 1 - \frac{\sum \mathbf{1}{\mathbf{G}=0}}{|\mathbf{G}|}$
> >
> > In this case, $\mathbf{I}$ quantifies the architecture differences between two concecutive epochs. Similarly, when $\mathbf{I}$ less than the threshold $\tau$, it also represents the consistent encoding of two contrastive paths with shared encoders.
> >
> > In summary, $\tau$ and $\Delta_s$ are orthogonal parameters designed for different types of SSL frameworks.
> >
> > - CSI activates the sparsity increment when I equals to $\Delta_s$, and this moment is defined as the CSI checkpoint", "the sparsification process is activated when I is less than a pre-defined threshold $\tau$ (e.g., $\tau$=0.9). Reproducibility could be affected by some parts that could use further explanation.
> >
> >   - **Response: ** Regarding the reproducibility, we evaluate the impact of the $\Delta_s$ in the response of \textbf{Q2} above. We added Table 9 in the Appendix of the revised manuscript. As shown in Figure 3 of the original manuscript, the architecture inconsistency always gradually decrease towards $\Delta_s$ during the early stage of training.  In Table 4 of the original manuscript, we demonstrated the performance of the proposed algorithm by running each experiment three times. The consistent superiority of the proposed algorithm indicates the minimum impact of the tunable parameter $\Delta_s$ and $\tau$.

---

> > > ### Comment · Reviewer_EBsm · 2022-11-30
> > > **comments on author response**
> > >
> > > Generally, response from the authors seems satisfactory.
> > >
> > > Eqs 10-12 now seem to heavily depend on (Liu et al, 2021) since some of the superscripts and subscripts are not defined.  However, "we propose the Synchronized Sparsification (SyncS)" and  "The proposed SyncS scheme gradually exploits the sparsity in both encoders while maintaining a consistent sparsity gap" are stated,  so Eqs 8-12 seem to part of the proposed SyncS.   I suggest to further clarify what you are proposing and what is borrowed from (Liu et al, 2021).

---

> > > > ### Author Response · Authors · 2022-12-10
> > > > **2nd round Response to Reviewer EBsm**
> > > >
> > > > **Q:** Generally, response from the authors seems satisfactory.
> > > >
> > > > Eqs 10-12 now seem to heavily depend on (Liu et al, 2021) since some of the superscripts and subscripts are not defined. However, "we propose the Synchronized Sparsification (SyncS)" and "The proposed SyncS scheme gradually exploits the sparsity in both encoders while maintaining a consistent sparsity gap" are stated, so Eqs 8-12 seem to part of the proposed SyncS. I suggest to further clarify what you are proposing and what is borrowed from (Liu et al, 2021).
> > > >
> > > > **A:** Thank you for your positive feedback on our previous responses.  To clarify the notations of Eq. 8 to Eq. 12, we created a detailed notation table as follow:
> > > > | Symbol       | Description                                                  |
> > > > | ------------ | ------------------------------------------------------------ |
> > > > | $f_{theta}$  | Online encoder                                               |
> > > > | $f_\xi$      | Offline (Momentum) encoder                                   |
> > > > | $g_\theta$   | Online projector                                             |
> > > > | $g_\xi$      | Offline (Momentum) projector                                 |
> > > > | $M_\theta$   | Weight masks of online encoder                               |
> > > > | $M_\xi$      | Weight masks of offline (momentum) encoder                   |
> > > > | $W_\theta$   | Weights of online encoder                                    |
> > > > | $W_\xi$      | Weights of offline (momentum) encoder                        |
> > > > | $s_\theta^t$ | Sparsity of the online encoder at iteration $t$              |
> > > > | $s_\theta^f$ | Target sparsity of the online encoder                        |
> > > > | $s_\xi^t$    | Sparsity of the offline (momentum) encoder at iteration $t$  |
> > > > | $s_\xi^f$    | Target sparsity of the offline (momentum) encoder            |
> > > > | $\Delta t$   | Sparsity update interval during the sparsity increment       |
> > > > | $\Delta_s$   | Sparsity gap (constraint) between online and offline encoder |
> > > > | $\tau$       | Architecture inconsistency threshold of the encoder          |
> > > >
> > > > We will add the above table to the appendix of the finalized manuscript (after the decision is made against our submission).
> > > >
> > > > We only adopted the sparsify scheduler from GraNet (Liu et al., 2021) in Eq. 10-12 for sparsity increment, and the encoders are sparsified by the proposed SyncCP and Contrastive Sparsification Indicator (CSI) methods. In the final version of the manuscript, we will further highlight the fact that the sparsity scheduler is the only component borrowed from GraNet (Liu et al., 2021).
> > > >
> > > > Thanks again for all your response and fruitful comments. We hope our detailed responses can resolve your questions and concerns, please kindly reconsider the overall score of our work.

---

### Official Review · Reviewer_NxVm · 2022-10-24

**Confidence:** 4
**Correctness:** 3
**Technical Novelty And Significance:** 4
**Empirical Novelty And Significance:** 4
**Recommendation:** 8

**Clarity, Quality, Novelty And Reproducibility:**

Novelty:
- The presented method is novel and interesting.

Clarity and Quality:
- It is a bit hard to understand what the authors are trying to do without making one pass through the paper. I would suggest the authors present their contributions and motivations more clearly/explicitly in the introduction. While I liked the gradual flow of the paper, I do want the spoilers right in the beginning.
- Table 7 does not indicate which dataset these numbers are for.
- The notation in equations 5, 8, and 9 makes it look like the output of the encoder is sparsified via multiplication with a mask. This does not apply any sparsity to the actual encoder parameters and thus does not bring the efficiency improvements the paper aims for.
- Typo: In paragraph above eq 7, "quantifying" -> "quantify"

Reproducibility:
- Hyperparameters and algorithms are presented in the Appendix.

**Strength And Weaknesses:**

Strengths:
- The paper walks you through the challenges of inducing sparsity during SSL training and then presents a method that addresses those challenges. I quite like the overall presentation.
- The proposed method is novel and well-motivated.

Weaknesses:
- The results for baseline BYOL on ImageNet with ResNet-50 (66.16) are very low compared to BYOL's reported numbers (74.3). Why?
- The presented method is only for momentum encoder-based SSL, which is not clear upfront. For instance, there is an introductory discussion about SimCLR and SDCLR, but the method is not actually applicable to SimCLR. Although the authors talk about generality to "other SSL methods," it is not clear which classes of SSL methods are included or excluded.
- Section 3 provides little context on the modeling decisions being made, such as why the online network is the one to be sparsified or why the online and offline encoders are pruned to different sparsity levels.

**Summary Of The Paper:**

- This paper explores the problem of inducing sparsity during training for self-supervised learning.
- The authors discuss challenges with applying existing sparse training techniques to self-supervised learning: sparsity-induced asymmetric non-identical encoders and sparsifying the model with frequent architectural changes both lead to degraded performance.
- The authors then present SyncCP, which sparsifies both encoders in an SSL setup while ensuring that the two encoders maintain a consistent gap between their sparsity levels and that their architectural difference is mainly due to this difference in sparsity levels rather than the disjointness of which weights to prune. The sparsity levels start gradually increasing once the architectural difference criterion is met.
- Experiments demonstrate improvements over other sparse training techniques and training speedups.

**Summary Of The Review:**

In general, I like this paper. It describes a problem worth solving and presents a method backed by a discussion of the challenges it aims to fix. However, I have some concerns with the current state of the paper (see Weaknesses and Clarity), which I believe need to be addressed before it can be ready for publication.

---

> ### Author Response · Authors · 2022-11-19
> **Response to Reviewer NxVM: Q1**
>
> **Q1:** The results for baseline BYOL on ImageNet with ResNet-50 (66.16) are very low compared to BYOL's reported numbers (74.3). Why?
>
> **A1:** Thank you for your response. In the original manuscript, we evaluated the proposed method with ResNet-50 on ImageNet based on BYOL framework. The lower accuracy is caused by the fact that, while the original BYOL paper trained the model with 1,000 epochs, we train the model from scratch with 200 epochs. In the revised manuscript, we increased the training effort with 300 epochs, and updated Table 6 with correspondingly higher accuracy for both baseline and sparse models (now the dense baseline accuracy is 68.12%). In the updated Table 6, we also added a footnote on the training epochs. Due to the extensive training time required for 1,000 epochs, we will report the experimental results with the 1,000-epoch training effort if our submission is accepted.
>
> Even with the reduced self-supervised pre-training effort with 300 epochs, the proposed method still achieves superior performance in the downstream task fine-tuning, and outperforms the conventional sparsification algorithms, as shown in the table below. Please note that the pre-trained sparse architectures are consistently preserved during the downstream fine-tuning.
>
> *[Table: Transfer learning performance of the ImageNet-trained BYOL encoder on CIFAR-10 and CIFAR-100 datasets.]*
>
> |                  Dataset                   |               | CIFAR-10 Acc (%) | CIFAR-10 Acc (%) | CIFAR-100 Acc (%) | CIFAR-100 Acc (%) |
> | :----------------------------------------: | :-----------: | :--------------: | :--------------: | :---------------: | :---------------: |
> |                  Encoder                   |               |    ResNet-50     |    ResNet-50     |     ResNet-50     |     ResNet-50     |
> |           Element-wise Sparsity            |               |        0%        |       80%        |        0%         |        90%        |
> |                                            |   SNIP [R1]   |      94.75       |      92.65       |       78.23       |       73.14       |
> |                                            |  GraSP [R2]   |      94.75       |      92.47       |       78.23       |       73.28       |
> |            Supervised Training             |   RigL [R3]   |      94.75       |      92.47       |       78.23       |       73.28       |
> |                                            |  GraNet [R4]  |      94.75       |      94.45       |       78.23       |       76.50       |
> | Transfer learning  from BYOL-trained model | **This work** |      96.09       |    **95.72**     |       80.13       |     **79.48**     |
>
> In other words, the proposed sparse self-supervised learning algorithm generates a sparse and powerful visual representation learner, achieving both high energy efficiency and outstanding performance in the downstream tasks. We added the table above in the Appendix of the revised manuscript as Table 8.

---

> > ### Author Response · Authors · 2022-11-19
> > **Response to Reviewer NxVM: Q2-Q3**
> >
> > **Q2:** The presented method is only for momentum encoder-based SSL, which is not clear upfront. For instance, there is an introductory discussion about SimCLR and SDCLR, but the method is not actually applicable to SimCLR. Although the authors talk about generality to "other SSL methods," it is not clear which classes of SSL methods are included or excluded.
> >
> > **A2:** We respectfully disagree with the reviewer’s comments. In the original manuscript, we reported experimental results with various SSL frameworks in Table 4 of the original manuscript, including both momentum encoder-based SSL (e.g., MoCoV2, BYOL) as well as shared encoderbased SSL (e.g., Barlow Twins). As clearly indicated in the original paper of Barlow Twins [R8] and open-sourced implementation [R9], the encoder is shared among two augmented inputs. Compared to SimCLR, Barlow Twins achieves much better accuracy with the same encoding scheme. Therefore, using Barlow Twins as the example of SSL (with shared encoder) is more reasonable than SimCLR. In the first paragraph of Section 5, we demonstrated the versatility of our proposed method that can be applied to both momentum encoder-based SSL and shared encoder-based SSL.
> >
> > [R8] Jure Zbontar, et al.,  "Barlow Twins: Self-supervised Learning via Redundancy Reduction." In International Conference on Machine Learning (ICML), 2021.
> >
> > [R9] D. Costa, et al. ``solo-learn: A Library of Self-supervised Methods for Visual Representation Learning.'' J. Mach. Learn. Res. 23 (2022): 56-1.
> >
> >
> >
> > **Q3:** Section 3 provides little context on the modeling decisions being made, such as why the online network is the one to be sparsified or why the online and offline encoders are pruned to different sparsity levels.
> >
> > **A3:** Thank you for your question. Contrastive self-supervised learning requires input augmentation which is further encoded by shared or separate encoders. Therefore, sparsifying both encoding paths during training maximizes the training efficiency, which motivates us to exploit weight sparsity in both encoders.
> >
> > As depicted in Figure 1(a), the previous self-damaging contrastive learning (SDCLR) demonstrated the effectiveness of the “sparse-dense” asymmetric SSL, i.e., encoding one of the augmentations with a highly sparse encoder (with fixed sparsity), while employing the dense encoding in the contrastive path. Section 3.1 empirically indicated the limitations of such “sparse-dense” contrastive learning. By default, the post-training linear evaluation is performed based on the trained online encoder. Therefore, we choose to sparsify the online encoder in Table 1 of the original manuscript.
> >
> > To fully address your question, we mirrored the experiment of Table 1 by exploiting high sparsity in the momentum encoder while keeping the online encoder dense:
> >
> > |      ResNet-18       |                | Dense Model Acc = 92.09% |                |               |
> > | :------------------: | :------------: | :----------------------: | :------------: | :-----------: |
> > |       Encoder        |     Online     |         Momentum         |     Online     |   Momentum    |
> > |    Fixed Sparsity    |      90%       |            0%            |      50%       |      0%       |
> > | Linear Eval. Acc (%) | 88.72 (-3.41%) |      87.68 (-4.31%)      | 92.10 (+0.01%) | 92.07(-0.02%) |
> > |    Fixed Sparsity    |       0%       |           90%            |       0%       |      50%      |
> > | Linear Eval. Acc (%) | 88.44 (-3.65%) |      87.52 (-4.57%)      |       -        |       -       |
> >
> > Following the same setting as SDCLR, sparsifying online (momentum) encoder while keeping the momentum (online) encoder dense shows largely degraded performance, as we demonstrated in Section 3.1 of the original manuscript.
> >
> > With the proposed CSI + SyncS sparsification method, we empirically observe that exploiting high sparsity in online model leads to better performance with gradient-based model update:
> >
> > *[Table: Performance comparison of BYOL on CIFAR-10 dataset with the proposed SyncS + CSI algorithm]*
> >
> > |     Method      | Online Encoder Spars. | Momentum Encoder Spars. | Online Linear Eval. Acc. (%) |
> > | :-------------: | :-------------------: | :---------------------: | :--------------------------: |
> > | **CSI + SyncS** |  0%$\rightarrow$50%   |   30%$\rightarrow$80%   |            91.88             |
> > | **CSI + SyncS** |  30%$\rightarrow$80%  |   0%$\rightarrow$50%    |          **92.24**           |
> >
> > Overall, sparsifying the online encoder leads to the optimal performance. we added the above tables in the Appendix of the revised manuscript as Table 11 and Table 12.

---

> > > ### Author Response · Authors · 2022-11-19
> > > **Response to Reviewer NxVM: Clarity and Quality**
> > >
> > > **Clarity and Quality**
> > >
> > > - It is a bit hard to understand what the authors are trying to do without making one pass through the paper. I would suggest the authors present their contributions and motivations more clearly/explicitly in the introduction. While I liked the gradual flow of the paper, I do want the spoilers right in the beginning.
> > >   - **Response:** In the revised manuscript, we updated Section I by presenting our contributions more clearly and explicitly.
> > > - Table 7 does not indicate which dataset these numbers are for.
> > >   - **Response:** Thank you for pointing this out, we updated the caption of Table 7 in the revised manuscript.
> > > - The notation in equations 5, 8, and 9 makes it look like the output of the encoder is sparsified via multiplication with a mask. This does not apply any sparsity to the actual encoder parameters and thus does not bring the efficiency improvements the paper aims for.
> > >   - **Response:** In the revised manuscript, we update the expression of Eq. (5), (8), and (9) with the dedicated weight masks for both online and offline encoders.
> > > - Typo: In paragraph above eq 7, "quantifying" $\rightarrow$ "quantify"
> > >   - **Response:** Thank you for pointing this out, we fixed the typo in the revised manuscript.

---

> > > ### Comment · Reviewer_NxVm · 2022-11-25
> > > **Thank you and follow-up question**
> > >
> > > Thank you for taking the time to respond to my comments.
> > >
> > > I think it is important to reproduce baseline numbers that align with previously published results, and I hope the authors are able to run longer experiments to get these numbers for camera-ready if the paper gets accepted.
> > >
> > > I understand that the method is applicable to both momentum-encoder-based SSL and shared-encoder-based SSL. I appreciate changing the title to specify contrastive SSL, but the text of the paper (for instance, the introduction and contributions) needs to be updated accordingly as well to clarify which classes of SSL the method is for.
> > >
> > > Regarding both encoder paths being pruned to different sparsity levels, can you please explain why they have to be different at all?

---

> > > > ### Author Response · Authors · 2022-12-10
> > > > **2nd round Response to Reviewer NxVm**
> > > >
> > > > **Q1:** I think it is important to reproduce baseline numbers that align with previously published results, and I hope the authors are able to run longer experiments to get these numbers for camera-ready if the paper gets accepted.
> > > >
> > > > **A1:** Thank you for your response.
> > > > We agree with you that the comparable baseline accuracy is important. We are currently running the experiments with longer training effort. The updated ImageNet accuracy will be ready for the camera-ready version (if the paper is accepted eventually).
> > > >
> > > > **Q2:** I understand that the method is applicable to both momentum-encoder-based SSL and shared-encoder-based SSL. I appreciate changing the title to specify contrastive SSL, but the text of the paper (for instance, the introduction and contributions) needs to be updated accordingly as well to clarify which classes of SSL the method is for.
> > > >
> > > > **A2:** Thank you for your feedback and suggestion. To address your question, we will clarify applicable SSL frameworks in the introduction section of the final manuscript.
> > > >
> > > > **Q3:** Regarding both encoder paths being pruned to different sparsity levels, can you please explain why they have to be different at all?
> > > >
> > > > **A3:** Thank you for your question.
> > > > The success of the current self-supervised learning is achieved by encouraging the encoders (e.g., CNN, ViT) to learn the asymmetric or decorrelated visual information.
> > > > In this work, the differently-sparsified encoder paths will generate the asymmetrically-encoded latent information between the higher sparsity encoder and the lower sparsity encoder. Minimizing the resultant contrastive loss or distance is equivalent to overcoming the sparsity-induced asymmetry, which further leads to both elevated efficiency and retained performance.
> > > > On the other hand, unifying the sparsity level between the two encoder paths is equivalent to removing the sparsity-induced asymmetry of SSL, while also weakening the learning capacity of both encoders.

---

### Official Review · Reviewer_inij · 2022-10-30

**Confidence:** 3
**Clarity, Quality, Novelty And Reproducibility:** I do not think I can reproduce this w…
**Correctness:** 4
**Technical Novelty And Significance:** 3
**Empirical Novelty And Significance:** 3
**Recommendation:** 5

**Strength And Weaknesses:**

**Strength**
1. The overall idea is novel and interesting.
2. The theoretical analysis of this work is promising.

**Weaknesses**

I am worried about the experimental results of this paper., although the performance of this work surpasses the prior work.

1. Obviously, in table 6, the linear probing performance is significantly lower when the training flops are reduced to 0.64× or 0.58× (64.89 vs. 66,16 and 63.76 vs. 66.16). I am wondering, in the case of reducing how many flops, this work can get the on-pair performance with the default setting,

2. Self-supervised learning aims to transfer the learned representations or whole network parameters into various downstream tasks. However, I do not see any transfer learning experiments in this paper. Could you provide more transfer learning experiments, for example, linear evaluation and fine-tuning in fine-grained classification tasks, semi-supervised learning, and object detection/segmentation?

**Summary Of The Paper:**

This paper proposes a new self-supervised learning framework (SSL), which aims to transfer the supervised sparse learning to SSL and reduce the required computational resources during the pretraining stage. Specifically, this paper investigates the correlation between training sparsity and SSL, which embraces the benefits of contrastiveness while exploiting high sparsity during SSL training. Experiments results show that this work surpasses the prior work on multiple datasets.

**Summary Of The Review:**

See Strength and Weaknesses above

---

> ### Author Response · Authors · 2022-11-19
> **Response to Reviewer IniJ: Q1**
>
> **Q1**: Obviously, in table 6, the linear probing performance is significantly lower when the training flops are reduced to 0.64× or 0.58× (64.89 vs. 66,16 and 63.76 vs. 66.16). I am wondering, in the case of reducing how many flops, this work can get the on-pair performance with the default setting.
>
> **A1:** Thank you for your question. In the updated Table 6 of the revised manuscript, we reported the results with the extended training effort 300 epochs, where the resultant models show 1.10% and 2.45% linear evaluation (probing) accuracy degradation with 80% and 90% sparsity, compared to dense baseline model. Since the 80% and 90% sparsity values are widely reported in recent supervised sparsification methods [R1, R3, R4], we would like to highlight that the proposed sparse self-supervised learning method achieves similar accuracy-sparsity tradeoff as the SoTA supervised sparsification counterpart, as shown in the table below:
>
> *[Table: Performance comparison between supervised and self-supervised sparse training algorithm]*
>
> |                Dataset                 |             | ImageNet-2012 Acc. Degradation (%) |                       |
> | :------------------------------------: | :---------: | :--------------------------------: | :-------------------: |
> |                Encoder                 |             |       ResNet-50 (1$\times$)        | ResNet-50 (1$\times$) |
> |         Element-wise Sparsity          |             |                80%                 |          90%          |
> |                                        |  SNIP [R1]  |            72.0 (-4.8)             |      67.2 (-9.6)      |
> |  Supervised Training Baseline = 76.8%  |  RigL [R3]  |            75.1 (-1.7)             |      73.0 (-3.8)      |
> |                                        | GraNet [R4] |            76.0 (-0.8)             |      74.5 (-2.3)      |
> | SSL Linear Eval. Acc  Baseline = 68.12 |  This work  |           67.02 (-1.10)            |     65.67 (-2.25)     |
>
> The following table shows the ImageNet linear evaluation accuracy with different sparsity. The proposed algorithm achieves the on-par or even better performance as the baseline with 50% target sparsity.
>
> |    Dataset     |                    |    ImageNet-2012     |                      |
> | :------------: | :----------------: | :------------------: | :------------------: |
> |    Sparsity    | Top-1 Accuracy (%) |   FLOPs (Training)   |     FLOPs (Test)     |
> | Dense Baseline |       68.12        | 7.94e+18 (1$\times$) | 8.18e+09 (1$\times$) |
> |      50%       |       68.31        |     0.78$\times$     |     0.68$\times$     |
> |      80%       |       67.02        |     0.62$\times$     |     0.33$\times$     |
> |      90%       |       65.67        |     0.56$\times$     |     0.22$\times$     |
>
> We added the above table to the Appendix of the revised manuscript as Table 17.

---

> > ### Author Response · Authors · 2022-11-19
> > **Response to Reviewer IniJ: Q2**
> >
> > **Q2:** Self-supervised learning aims to transfer the learned representations or whole network parameters into various downstream tasks. However, I do not see any transfer learning experiments in this paper. Could you provide more transfer learning experiments, for example, linear evaluation and fine-tuning in fine-grained classification tasks, semi-supervised learning, and object detection/segmentation?
> >
> > **A2:** Thank you for your comment on this. Based on the reviewer’s feedback, we performed additional experiments and verified the sparse ImageNet-trained model with the downstream tasks. The following table summarizes the transfer learning performance by fine-tuning the ImageNet-trained sparse BYOL encoder on the downstream CIFAR-10 and CIFAR-100 datasets.
> >
> > *[Table: Transfer learning performance of the ImageNet-trained BYOL encoder on CIFAR-10 and CIFAR-100 datasets.]*
> >
> > |                  Dataset                   |               | CIFAR-10 Acc (%) | CIFAR-10 Acc (%) | CIFAR-100 Acc (%) | CIFAR-100 Acc (%) |
> > | :----------------------------------------: | :-----------: | :--------------: | :--------------: | :---------------: | :---------------: |
> > |                  Encoder                   |               |    ResNet-50     |    ResNet-50     |     ResNet-50     |     ResNet-50     |
> > |           Element-wise Sparsity            |               |        0%        |       80%        |        0%         |        90%        |
> > |                                            |   SNIP [R1]   |      94.75       |      92.65       |       78.23       |       73.14       |
> > |                                            |  GraSP [R2]   |      94.75       |      92.47       |       78.23       |       73.28       |
> > |            Supervised Training             |   RigL [R3]   |      94.75       |      92.47       |       78.23       |       73.28       |
> > |                                            |  GraNet [R4]  |      94.75       |      94.45       |       78.23       |       76.50       |
> > | Transfer learning  from BYOL-trained model | **This work** |      96.09       |    **95.72**     |       80.13       |     **79.48**     |
> >
> > Please note that the pre-trained sparse architectures are consistently preserved during the downstream fine-tuning. As shown in the table above, the strong visual representation learners are obtained by training the backbone model with self-supervised learning on ImageNet dataset. Sparsified by the proposed algorithm on ImageNet with 300 epochs during pre-training, fine-tuning the SSLtrained backbone model leads to 1.08% and 1.59% accuracy improvements compared to the recent SoTA supervised pruning algorithms. We will investigate the semi-supervised learning tasks in future works.
> >
> > In the Appendix of the revised manuscript, we added the table above (as Table 8) and corresponding descriptions on downstream transfer learning tasks.

---

> > > ### Comment · Reviewer_inij · 2022-11-23
> > > **Some further questions about the baseline**
> > >
> > > Thanks for your efforts in addressing my concerns. However, I have some further questions regarding to the baseline results in this paper.
> > >
> > > 1. The baseline result of BYOL in this paper is too low. I understand that using a small batch size might reduce the performance. However, some previous works reproduced much higher numbers than this paper. For example, [1] reported 71.4% linear probing accuracy with a batch size of 512 and 200 training epochs, [2] reported  69.6% / 71.8% with a batch size of  128 / 256 for 300 training epochs.
> > >
> > > 2. The transfer learning baseline is too low. From table 3 in BYOL paper (last row),  we can achieve 95.9% and 80.2% Top-1 on CIFAR-10 and CIFAR-100 **even without any pretrain**
> > >
> > > 3. The transfer learning experiments are too weak, I can understand that maybe the author does not have enough computing resources to conduct detection and segmentation experiments. However, even for the classification experiments, using only CIFAR-10 and CIFAR-100 is not convincing. Please report more experimental results on different datasets (e.g. Table 3 in BYOL paper).
> > >
> > > [1] Exponential Moving Average Normalization for Self-supervised and Semi-supervised Learning.
> > >
> > > [2] Momentum2 Teacher: Momentum Teacher with Momentum Statistics for Self-Supervised Learning.

---

> > > > ### Author Response · Authors · 2022-12-10
> > > > **2nd round Response to Reviewer inij**
> > > >
> > > > **Q:** Thanks for your efforts in addressing my concerns. However, I have some further questions regarding to the baseline results in this paper.
> > > >
> > > > **A:** Thank you for raising the questions regarding the baseline accuracy.
> > > >
> > > > **Q1:** The baseline result of BYOL in this paper is too low. I understand that using a small batch size might reduce the performance. However, some previous works reproduced much higher numbers than this paper. For example, [1] reported 71.4% linear probing accuracy with a batch size of 512 and 200 training epochs, [2] reported 69.6% / 71.8% with a batch size of 128 / 256 for 300 training epochs.
> > > >
> > > > **A1:**  To address your concern, we re-run the pre-training experiments on ImageNet dataset with BYOL framework and 300 epochs. We increased the batch size from 128 to 256. Following [1] referred by the reviewer, we set the initial momentum of EMA to 0.98, then gradually increased the momentum to 0.999. With the updated parameters, we obtained improved performance of both dense baseline and 80% sparse models, as reported in the table below:
> > > >
> > > > | ImageNet       | Model     | Linear Eval. Acc |
> > > > | -------------- | --------- | ---------------- |
> > > > | Dense Baseline | ResNet-50 | 70.42            |
> > > > | 80% sparse     | ResNet-50 | 69.19            |
> > > >
> > > >
> > > >
> > > > **Q2:** The transfer learning baseline is too low. From table 3 in BYOL paper (last row), we can achieve 95.9% and 80.2% Top-1 on CIFAR-10 and CIFAR-100 even without any pretrain.
> > > >
> > > > **Q3:** The transfer learning experiments are too weak, I can understand that maybe the author does not have enough computing resources to conduct detection and segmentation experiments. However, even for the classification experiments, using only CIFAR-10 and CIFAR-100 is not convincing. Please report more experimental results on different datasets (e.g. Table 3 in BYOL paper).
> > > >
> > > > [1] Exponential Moving Average Normalization for Self-supervised and Semi-supervised Learning.
> > > >
> > > > [2] Momentum2 Teacher: Momentum Teacher with Momentum Statistics for Self-Supervised Learning.
> > > >
> > > >
> > > >
> > > > **A2 & A3:** With the improved pre-trained model improvements, we further validated the trained encoder with a number of additional downstream tasks on multiple datasets besides CIFAR-10 and CIFAR-100, now including Food101, Cars, Flowers, CalTech101, DTD, Pets, VOC2007, and Aircraft. This is similar to Table 3 of the BYOL paper, as the reviewer suggested. These additional results on downstream tasks are summarized in the table below, for the dense model and the 80% sparse model.
> > > >
> > > > |            | CIFAR10 | CIFAR100 | Food101 | Cars  | Flowers | CalTech101 |  DTD  | Pets  | VOC2007 | Aircraft |
> > > > | :--------: | :-----: | :------: | :-----: | :---: | :-----: | :--------: | :---: | :---: | :-----: | :------: |
> > > > |   Dense    |  96.64  |  82.23   |  78.90  | 87.31 |  94.31  |   91.78    | 73.01 | 90.10 |  82.63  |  76.98   |
> > > > | 80% Sparse |  96.15  |  81.52   |  78.31  | 87.02 |  94.18  |   91.47    | 72.37 | 90.86 |  82.31  |  76.49   |
> > > >
> > > > To address your concern regarding the comprehensive downstream tasks, }we followed the open-sourced SSL transfer learning implementation [(official code)](https://github.com/linusericsson/ssl-transfer) of the CVPR 2021 paper [R1]. We finetuned the models for 5,000 steps with SGD and batch size of 64.
> > > > Following the setup in [R1], the initial learning rate is chosen from a grid of 4 logarithmically spaced values between 0.0001 and 0.1. The weight decay is similarly chosen from a grid of 4 logarithmically spaced values between 1e-6 and 1e-3. For each hyperparameter setup inside the sweeping range, we re-load the pretrained backbone model. In other words, the experimental results between different training setup are independent and we choose the best accuracy as the reported results. With the re-trained backbone model, the finetuned backbone model achieves a comparable performance as the reported BYOL downstream performance in [R1].With the improved SSL-trained baseline model, the accuracy of the downstream tasks outperforms the accuracy with random initialization that was presented in the BYOL paper, as shown in the table above. Admittedly the 300-epoch pre-training cannot fully duplicate the 1000-epoch trained results reported in the BYOL paper, but }as we responded to Reviewer NxVM, we will update all ImageNet results with full-sized 1,000-epoch training, and we will also include the updated comprehensive downstream task results in the final version of the manuscript (if our paper is accepted eventually).
> > > >
> > > > [R1] E, Linus, et al., "How well do self-supervised models transfer?." Proceedings of the IEEE/CVF Conference on Computer Vision and Pattern Recognition (CVPR). 2021.

---

### Official Review · Reviewer_19nX · 2022-10-31

**Confidence:** 3
**Clarity, Quality, Novelty And Reproducibility:** 1. The paper proposed a novel synchro…
**Correctness:** 3
**Technical Novelty And Significance:** 2
**Empirical Novelty And Significance:** 2
**Recommendation:** 3

**Strength And Weaknesses:**

Strength
1. The paper presented a detailed analysis of the existing sparse self-supervised learning approaches.
2. The proposed approach is novel in solving the sparsity in SSL and also improve the model performance.

Weakness
1.The paper did not give a sufficient introduction of its baseline, SDCLR. It is hard to follow the ideas without that background.
2. The paper did not evaluate the inference complexity, e.g. inference FLOPs, latency, memory cost, etc. Without that, it is hard to justify the effectiveness of its sparsity.
3. The proposed approach is contrastive learning focused. If the paper is scoped to the contrastive SSL, the title needs to be confined. Otherwise, the paper needs to further evaluate its performance on MIM SSL frameworks.


**Summary Of The Paper:**

This paper proposed a new sparse SSL approach by exploring the correlation between in-training sparsity and SSL. This paper investigates the challenges of the sparsity-induced asymmetric SSL (a.k.a prune-and-regrow) and proposes synchronized contrastive pruning approach. The proposed approach is adaptive to various granularities of sparsity.


**Summary Of The Review:**

The paper presented a novel synchronized contrastive pruning approach in sparse SSL. As its main contribution, the work did not evaluate the inference efficiency which is hard to justify the effectiveness of its sparsity. The paper can be further improved by introducing the core idea of its baseline to make the paper easy to follow.

---

> ### Author Response · Authors · 2022-11-19
> **Response to Reviewer 19nX**
>
> **Q1:** The paper did not give a sufficient introduction of its baseline, SDCLR. It is hard to follow the ideas without that background.
>
> **A1:** Thank you for your response. In the original manuscript of the paper, we attempted to introduce the characteristics of the self-damaging contrastive learning (SDCLR) with Figure 1(a) and a paragraph in the Section I, and we further described the details and limitations of SDCLR in Section 3.1. Based on the reviewer’s feedback, we added additional descriptions of SDCLR in Section I of the updated manuscript, to enhance the understanding of the readers.
>
> **Q2:** The paper did not evaluate the inference complexity, e.g. inference FLOPs, latency, memory cost, etc. Without that, it is hard to justify the effectiveness of its sparsity.
>
> **A2:** In Table 5 and Table 6 of the original manuscript, we presented the FLOPs reduction and inference speedup on GPU with CIFAR-10 dataset.
> To address your question, we report the inference FLOPs reduction in Table 6 of the revised manuscript. With 90% sparsity, the proposed algorithm achieves up to 0.22$\times$ inference FLOPs reduction. In both of Table 5 and Table 6 of the revised manuscript, we add the total memory consumption of the sparsified ResNet-18 and ResNet-50 model on the CIFAR and ImageNet datasets, respectively. Specifically, we also include the baseline inference time in Table 5 for your reference.
>
> **Q3:** The proposed approach is contrastive learning focused. If the paper is scoped to the contrastive SSL, the title needs to be confined. Otherwise, the paper needs to further evaluate its performance on MIM SSL frameworks.
>
> **A3:** To avoid such potential confusion, we revised the title to "Synchronized Pruning for Efficient Contrastive Self-Supervised Learning'',
> which emphasizes our focus on contrastive SSL rather than MIM-type SSL.
>
> **Clarity and Quality:**
>
> - The paper proposed a novel synchronized contrastive pruning approach to solve the problems of the existing sparse SSL approaches.
> - The paper is clearly presented in general. However, given the paper is an improvement of the existing SDCLR work, missing a brief introduction of the core framework of SDCLR makes the paper hard to follow.
>   - **Response: ** We believe we have provided sufficient information of SDCLR in introduction and later sections. To enhance your understanding, we added additional details of SDCLR in the introduction section of the revised manuscript.
> - The paper provided the details about the experiments. A minor issue is it did not clearly present the dataset used for the pretraining and linear probe.
>   - **Response:** We described the detailed experimental setups in Section 5 and the Appendix. The names of the datasets are indicated in the captions of Table 4, Table 5, and Table 6. Based on the reviewer's feedback, we added the dataset information for Table 7 in the updated manuscript.

---

> ### Author Response · Authors · 2022-12-04
> **Looking forward to your feedback**
>
> Dear reviewer 19nX,
>
> Thanks again for your time. As the deadline for discussion is approaching, we really hope to have a further discussion with you to see if our response solves the concerns. We are happy to provide any additional clarifications that you may need.
>
> Best regards, Author

---

### Official Review · Reviewer_5j3N · 2022-10-31

**Confidence:** 4
**Clarity, Quality, Novelty And Reproducibility:** Na
**Correctness:** 3
**Technical Novelty And Significance:** 2
**Empirical Novelty And Significance:** 2
**Recommendation:** 5

**Strength And Weaknesses:**

Strength:

The paper is well written.

The experiments including various different sparse types (unstructured, N:M, structured) are persuasive.

The performance (accuracy and efficiency) improvement is significant compared to the existing method [1].




Weaknesses:

The self-supervised learning method usually trains on large-scale datasets, but this paper doesn't show any training GPU hours improvement on large-scale datasets. I feel confused about the motivation.

The NVidia GPUs only support 2:4 sparsity, can you explain the detail implementation of inference gain about 1:4 sparsity in Table5.


The BYOL and MOCO-v2 are not SOTA SSL methods, the recent SOTA methods are more convincing.

**Summary Of The Paper:**

This paper enables model pruning to accelearate contrastive self-supervised training. Compared to the exsited method, this paper tries to sparsify online decoder and offline decoder simultaneously. To stabilize the sparse training. The authors propose Contrastive Sparsification Indicator (CSI) to guide the model pruning.

To evaluate the effectiveness of SyncCP, the authors conduct experiments on various classification datasets with unstructured sparsity, N:M sparsity, and structured pruning.

**Summary Of The Review:**

Na

---

> ### Author Response · Authors · 2022-11-19
> **Response to Reviewer 5j3N**
>
> **Q1:** The self-supervised learning method usually trains on large-scale datasets, but this paper doesn't show any training GPU hours improvement on large-scale datasets. I feel confused about the motivation.
>
> **A1:** We thank you for your review. To the best of our knowledge, directly achieving the GPU training acceleration is unfortunately infeasible, due to the fact that the sparsity operation of backward introduced by current sparse training methods [R1-R4] is not supported well in Nvidia GPUs. For example, although the recently proposed N:M sparsity can accelerate the computation in the forward pass, the backward gradient computation remains as dense computing. Similar to the prior SoTA works [R1-R4], we report the training FLOPS reduction in Table 6 of the original manuscript. Furthermore, we measure the actual training speedup based on custom DNN accelerator chip from the literature with dedicated structured sparsity, as summarized in Table 7 of the original manuscript.
>
> **Q2:** The NVidia GPUs only support 2:4 sparsity, can you explain the detailed implementation of inference gain about 1:4 sparsity in Table 5?
>
> Thank you for your question. In this work, we adopted the Nvidia-Apex library to accelerate the inference process with N:M sparsity.
> Overall, the Apex library exploits the N:M sparsity based on the following steps:
>
> - Initialize the pruner by defining the mask calculator [official Apex code](https://github.com/NVIDIA/apex/blob/9866160066a024f848f4ce4eb9aabc7ce796ec50/apex/contrib/sparsity/asp.py#L301). The sparsity is exploited based on the pre-defined pattern.
> - By default, the sparsity pattern is 2:4, where the N and M values are **hard-coded** into the general pruning function [official Apex code](https://github.com/NVIDIA/apex/blob/9866160066a024f848f4ce4eb9aabc7ce796ec50/apex/contrib/sparsity/sparse_masklib.py#L50).
> - As long as the group size $M$ is a multiple of 4, the customized sparsity can be supported by modifying the hard-coded sparse pattern in the general pruning function.
>
> For example, the N:M=1:4 sparsity can be formulated by specifying M=4, N=1 in the general pruning function.  Subsequently, the stop-level sparsity enabler will activate 75% overall sparsity with the printed out message.
>
> On the algorithm level, we exploit the N:M sparsity based on the same sparsification scheme as the Apex library. In the Apex library, the top level "sparsity enabler'' will assign the intrinsic sparsity mask to each layer [official Apex code](https://github.com/NVIDIA/apex/blob/9866160066a024f848f4ce4eb9aabc7ce796ec50/apex/contrib/sparsity/asp.py#L154), and the aforementioned pruning function will update the mask accordingly based on the user-defined sparsity pattern.
>
> **Q3:** The BYOL and MOCO-v2 are not SOTA SSL methods, the recent SOTA methods are more convincing.
>
> **A3:** We respectively disagree that the recent SOTA methods are more convincing for verifying the effectiveness of our scheme, than BYOL and MOCO-v2. We first emphasize that our focus is on contrastive SSL methods rather than transformer-based SSL such as MAE and BEIT. We think BYOL and MOCO are not only the most representative contrastive SSL methods not-using and using negative pairs of view, respectively, but also still comparable to SOTA SSL methods. To the best of our knowledge, the SOTA ImageNet linear evaluation result among all contrastive SSL methods is 81.0% which is achieved by a variant of MOCO-v2 (i.e., MOCO-v3) [R5]. Next, the best result achieved by BYOL is 79.6% [R6] which is also comparable with the SOTA result. Here, as far as we know, ReLICv2 and SimCLRv2 are only two contrastive SSL methods (other than MOCO-v3) which report better accuracy than 79.6%.
>
> Furthermore, we would like to highlight that, as presented in the original manuscript, one of the major objectives of this paper is to demonstrate the incompatibility of the supervised sparse training method in SSL and further illustrate the limitations of the recent self-damaging SSL. We choose the CNN-based SSL as the platform while the transformer-based sparse SSL [R7] is left as our future work.
>
> [R1] Namhoon Lee, et al., SNIP: Single-shot Network Pruning based on Connection Sensitivity. ICLR, 2018.
>
> [R2] C. Wang, et al., "Picking winning tickets before training by preserving gradient flow." ICLR, 2020.
>
> [R3] U. Evci, et al. "Rigging the lottery: Making all tickets winners." PMLR, 2020.
>
> [R4] Shiwei Liu, et al. "Sparse Training via Boosting Pruning Plasticity with Neuroregeneration". NeurIPS, 2021.
>
> [R5] Xi. Chen, et al. "An Empirical Study of Training Self-Supervised Vision Transformers." ICCV. 2021.
>
> [R6] J-B. Grill, et al. "Bootstrap your own latent: A new approach to self-supervised Learning." NeurIPS. 2020.
>
> [R7] He, Kaiming, et al. "Masked autoencoders are scalable vision learners.'' CVPR. 2022.

---

> ### Author Response · Authors · 2022-12-04
> **Looking forward to your feedback**
>
> Dear reviewer 5j3N,
>
> Thanks again for your time. As the deadline for discussion is approaching, we really hope to have a further discussion with you to see if our response solves the concerns. We are happy to provide any additional clarifications that you may need.
>
> Best regards,
>
> Author

---

### Author Response · Authors · 2022-12-13
**Message to the Area Chair**

We would like to thank the area chair and all the reviewers for their great efforts during the review process. Here, we are writing to bring to your attention that we notice that reviewer 5J3N and reviewer 19nX has no responses to our rebuttal after two times friendly reminder.

(1) Reviewer 5j3N was mainly questioning the implementation with different N:M sparsity patterns other than 2:4, we provided a detailed explanation with the line-by-line instructions based on the official Nvidia Apex library.

(2) We would like to highlight that, as presented in the original manuscript, one of the major objectives of this paper is to demonstrate the incompatibility of the supervised sparse training method in SSL and further illustrate the limitations of the recent self-damaging SSL. As one of the first works in this area, the proposed method aims to elevate the energy efficiency of powerful self-supervised learning, and further leads to energy-efficient downstream applications.

(3) Reviewer 19nX is subjective about the clarity of writing and descriptions of experiments. We have carefully addressed his comments in both rebuttal and revision. To further address the reviewer's confusion regarding the MIM-based SSL, we revised the title to "Synchronized Pruning for Efficient Contrastive Self-Supervised Learning'', which emphasizes our focus on contrastive SSL rather than MIM-type SSL.

We will greatly appreciate it if the area chair can use your own expertise to judge the reviewer's comments.

Best regards,
Authors

---

### Decision · Program_Chairs · 2023-01-20

**Decision:**

Reject

**Justification For Why Not Higher Score:**

Important technical details shall be better clarified. For example,  Eqs 10-12 are borrowed from existing works, which somehow have not been clearly explained by the authors and will degrade the novelty of this paper. Also, reviewers have major concerns about the correctness of baseline performance, e.g., the lower baseline result of BYOL, and the lower transfer learning baseline. Also, the transfer learning experiments are weak. Though the authors updated some extra experimental results with the new training recipe at the last minute, obviously this is a significant issue, which may request a major revision of the whole experiment section in this paper. Also, reviewers suggested including more state-of-the-art comparison methods, rather than those classical methods.

**Justification For Why Not Lower Score:**

N/A

**Metareview: Summary, Strengths And Weaknesses:**

This paper studied self-supervised learning and aimed to transfer the supervised sparse learning to self-supervised learning so that the computational resources during the pretraining stage can be reduced. The idea of this paper is interesting and the authors also provided some relevant analyse, which is very nice. But the experimental part may not be solid enough to well justify the effectiveness of the proposed algorithm.